

**Diazotrophic *Trichodesmium* influence on ocean color and**
**pigment composition in the South West tropical Pacific**
**Cécile Dupouy[1*], Robert Frouin[2], Marc Tedetti[1], Morgane Maillard[1],**
**Martine Rodier[3], Fabien Lombard[4], Lionel Guidi[4], Marc Picheral[4], Solange**
**Duhamel[5], Bruno Charrière[6], Richard Sempéré[1]**
[1] Aix Marseille Univ, Université de Toulon, CNRS, IRD, MIO UM 110, 13288, Marseille, France
[2] Scripps Institution of Oceanography, University of California San Diego, La Jolla, California, USA
     [3] Environnement Insulaire Océanien (EIO), UMR 241 (Université de Polynésie Française, Institut de
     Recherche pour le Développement, Institut Louis Malardé, IFREMER), Centre IRD de Tahiti, BP 529,
     98713 Papeete, French Polynesia
[4] Sorbonne Universités, UPMC Université Paris 06, CNRS, Laboratoire d'Océanographie de
Villefranche (LOV), Observatoire Océanologique, 06230 Villefranche-sur-Mer, France
[5] Lamont Doherty Earth Observatory, Columbia University, Palisades, New York, USA
[6] CNRS, Cefrem, Université de Perpignan. 52, av. Paul Alduy 66860, Perpignan Cedex, France
* Aix Marseille Université, CNRS/INSU, Université de Toulon, IRD, Mediterranean Institute of
Oceanography (MIO) UM 110, 98848, Nouméa, New Caledonia

* Corresponding author: Aix Marseille Univ, Université de Toulon, CNRS, IRD, MIO UM
110, 13288, Marseille, France : cecile.dupouy@ird.fr
• **Keywords:** *Trichodesmium*, chlorophyll, pigments, normalized water leaving
radiances, inherent optical properties, South West tropical Pacific
**Abstract**
We assessed the influence of the marine diazotrophic cyanobacterium *Trichodesmium* on the
bio-optical properties of  South West tropical Pacific waters (18-22 °S, 160 °E-160 °W)
during the February-March 2015 OUTPACE cruise. We performed measurements of
backscattering and absorption coefficients, irradiance, and radiance, in the euphotic zone, and
took Underwater Vision Profiler 5 (UPV5) pictures for counting the largest *Trichodesmium*





spp colonies. Pigment concentrations were determined by fluorimetry and by high
performance liquid chromatography and picoplankton abundance by flow cytometry.
Trichome concentration was estimated from pigment algorithms and validated by surface
visual counts. In result, the large colonies were well correlated to the trichome concentration
estimates (though with a large factor of 600 to 900, due to aggregation processes). Large
*Trichodesmium* abundance was always associated with particulate absorption at a peak of
mycosporine-like amino acid absorption, and high particulate backscattering, but not with
high fluorescence, high chlorophyll-a concentration, or blue particulate absorption in the
water column. Along the West to East transect, *Trichodesmium* together with
*Prochlorococcus* represented the major part of total chlorophyll and the other groups were
negligible. *Trichodesmium* contribution to chlorophyll was the highest in the Melanesian
Archipelago around New Caledonia and Vanuatu, progressively decreased to the vicinity of
the Fiji Islands, and reached a minimum in the South Pacific gyre where the contribution of
*Prochlorococcus* was maximum. At the frontal LDB, *Trichodesmium* and *Prochlorococcus*
has almost same contributions. The relationship between normalized water-leaving radiance,
in the ultraviolet and visible domains, $nL_w$, and chlorophyll was generally similar to that
found in the Eastern tropical at BIOSOPE. Principal component analysis (PCA) of
OUTPACE data showed that $nL_w$ were strongly correlated to chlorophyll except in the green
and yellow domains. These results, as well as differences in the PCA of BIOSOPE data,
suggested that $nL_w$ variability in the green and yellow during OUTPACE was influenced by
other variables, associated with *Trichodesmium* presence as the backscattering coefficient,
phycoerythrin fluorescence, and/or zeaxanthin absorption. *Trichodesmium* detection should
then involve examination of $nL_w$ at the green and yellow wavelengths.

## 1 Introduction


The ecological importance of filamentous diazotrophs (*Trichodesmium* spp. in particular) in
the archipelago region of the South West tropical Pacific (SWTP) has long been pointed out
(Dandonneau and Nyang, 2007). *Trichodesmium* spp. have to be taken into account for the
estimation of the global oceanic nitrogen and carbon cycles (Capone and Carpenter, 1997;
Bonnet et al., 2017; Dutheil et al., this issue). In the past decade, efforts have been made to
extract abundances of different phytoplanktonic taxonomic groups from ocean color data
(Blondeau-Patissier et al., 2014; Bracher et al., 2017). Other attempts have been made to get
remote sensing estimates of the abundance and diazotroph activity of *Trichodesmium* at a
global scale (Westberry and Siegel, 2005; McKinna et al., 2011; Dupouy et al., 2011;
McKinna, 2015). Satellite detection of *Trichodesmium* is facilitated when concentration at the
sea surface is high, leading to a building of mat larger than a satellite pixel. These mats induce
a high reflectance in the near infrared, a "red edge", which can easily be observed (Hu et al.,



2010; Dupouy et al., 2011; McKinna et al., 2011; Gower et al., 2014; McKinna, 2015; Rousset
et al., this issue). Detection becomes more difficult when *Trichodesmium* concentrations are at
non-bloom or sub-bloom abundance, i.e. when colonies are distributed throughout the water
column and mixed with other species. Using empirical statistical approach, De Boissieu et al.
(2014) determined that at sufficient concentration level, these filamentous diazotrophs can be
distinguished from other groups. This complements empirical parameterizations that were
used to derive the vertical distribution of different phytoplankton groups (microplankton,
nanoplankton, and picoplankton) using High Performance Liquid Chromatography (HPLC)
diagnostic pigments and surface chlorophyll a (Chla) determination from space (Uitz et al.,
2006; Ras et al., 2008; Brewin et al., 2011).

In order to validate *Trichodesmium* discrimination algorithms, and to improve the

knowledge of the influence of *Trichodesmium* spp. on apparent (AOPs) and inherent (IOPs)
optical properties of seawater, accurate field determinations are required. E.g., it is necessary
to measure the normalized water-leaving radiance [$nL_w(\lambda)$ in W m$^{-2}$ sr$^{-1}$], i.e., the radiance that
emerge from the ocean in the absence of atmosphere, with the Sun at zenith, at the mean
Earth-Sun distance (Gordon, 2005). $nL_w(\lambda)$ is governed by two main IOPs (Mobley 1994;
Kirk, 1994): volume absorption [$a(\lambda)$ in m$^{-1}$] and volume backscattering [$b_b(\lambda)$ in m$^{-1}$]
coefficients. IOPs are controlled by the concentrations of optically active components in a
volume of water, which include phytoplankton and colored detrital matter (CDM), the latter
being composed by non algal particulate matter (NAP) and chromophoric dissolved organic
matter (CDOM). If AOPs are well related to phytoplankton pigments in Case I oceanic waters
(Morel and Maritorena, 2001, Morel et al., 2007), this relationship might be modified by the
presence of *Trichodesmium* (with moderate Chla concentrations < 1 mg m$^{-3}$). As summarized
in Westberry and Siegel (2005), *Trichodesmium* displays unique optical properties that may
allow their detection: (1) a strong absorption in the ultraviolet (UV) domain related to the
presence of mycosporin like amino-acids (Subramaniam et al., 1999a; Dupouy et al, 2008),
(2) a higher relative reflectance near 570 nm due to phycoerythrin fluorescence (Borstad et al.,
1992; Subramaniam et al., 1999b), and (3) an increased backscattering across all wavelengths
caused by the index of refraction change within intracellular gas vacuoles (Borstad et al.,
1992; Subramaniam et al., 1999b; Dupouy et al., 2008).
The SWTP between New Caledonia and the Tonga trench is particularly rich in
*Trichodesmium* colonies during summer (Dupouy et al., 1988; 2000; 2011; Biegala et al.,
2014) and this richness further enhanced during the positive phase of the ENSO in 2003





(Tenorio et al., in press). Using bio-optical measurements, this study aims (1) to describe
several AOPs and IOPs of interest in the UV and visible domains of SWTP waters, as well as
pigments, and abundance of all phytoplanktonic cells including large and smaller
*Trichodesmium* colonies and picoplankton, (2) to determine the influence of *Trichodesmium*
spp. on *in situ* measurements of ocean color, and absorption and backscattering coefficients.
For this purpose, we used identical measurements than those implemented in the tropical
oligotrophic ocean during the BIOSOPE cruise (Tedetti et al., 2007; 2010).

**2 Material and methods**

**2.1 Study area**

The "Oligotrophy from Ultra-oligoTrophy PACific Experiment (OUTPACE)" cruise

was conducted on board the RV L'Atalante from 21 February  to 31 March 2015 in the South
West tropical Pacific Ocean (Table 1; Fig. 1). *In situ* measurements and water sampling were
performed at fifteen stations along a 4000-km transect extending from the mesotrophic waters
of the Melanesian Archipelago (MA, SD1 to SD6), near New Caledonia and Vanuatu to the
Fijian archipelago (FI, SD7 to SD12), between Fiji and Tonga, and to the extreme eastern end
in the hyper-oligotrophic waters of the South Pacific East of Tonga Trench in the Gyre (SPG,
SD12 to SD15). General biogeochemical and hydrographic characteristics of the waters along
this transect are described in Moutin et al. (this issue).

**2.2. Radiometric measurements and determination of $nL_w(\lambda)$, $K_d(\lambda)$ and $Z_{10\%}(\lambda)$ values**

At each station, two or more profiles of downward irradiance [$E_d(Z, \lambda)$ in µW cm$^{-2}$

nm$^{-1}$] and upward radiance [$L_u(Z, \lambda)$ in µW cm$^{-2}$ nm$^{-1}$ sr$^{-1}$] were made at each station around
solar noon using a Satlantic MicroPro free-falling profiler equipped with OCR-504 downward
irradiance and upward radiance sensors with UV-B (305 nm), UV-A (325, 340 and 380 nm)
and visible (412, 443, 490 and 565 nm) spectral channel, as further described in Tedetti et al.
(2010). The MicroPro free-fall profiler was operated from the rear of the ship and deployed
30 m away to minimize the disturbances of the ship. Surface irradiance [$E_s(\lambda)$ in µW cm$^{-2}$ nm$^{-}$
$^{1}$] was concomitantly measured at the same wavelengths on the ship deck using other OCR-
504 sensors to account for the variations of cloud conditions during the cast. Satlantic, Inc.
surface and in-water radiometers were calibrated before the cruise. Mostly cloudy sky
conditions existed during the profiles (only a few acquisitions were made under clear skies),





and at SD5 at 17:30-19:00 they were made under a heavy shower. SD3, SD4, and SD13
profiles were not available (night stations). Details of the casts can be found in Appendix A.
Determination of $nL_w(\lambda)$ was conducted from values of $L_u(Z, \lambda)$, diffuse attenuation
coefficient for upward radiance ($K_L(\lambda)$ in m$^{-1}$), water-leaving radiance [$L_w(Z, \lambda)$] and $E_s(\lambda)$
(see calculations in Appendix A). The $nL_w(\lambda)$ data presented in this study are average values
of two to three upward radiance casts (coefficient of variation < 8% for all the stations
concerned). Diffuse attenuation coefficient for downward irradiance ($K_d(\lambda)$ in m$^{-1}$) was
determined using $E_d(Z, \lambda)$ and $E_s(\lambda)$ values (Appendix A).
The first optical depth corresponding to the surface layer observed by the satellite
ocean color instruments (Kirk, 1994) [$Z_{10\%}(\lambda)$ in m] was extrapolated from $K_d(\lambda)$ and
calculated as $\ln(10)/K_d(\lambda)$. In this study, the integrated concentrations of the different
microorganisms between the surface and the first optical depth were used to determine the
relationship between these concentrations and $nL_w(\lambda)$ values.

**2.3 Water sampling**
Seawater samples were collected during the noon cast of each station at different
depths using 12-L Niskin bottles for the determination of various parameters. For the
determination of Chla concentration and particulate (phytoplankton + NAP) absorption
coefficient [$a_P(\lambda)$], samples were collected at depths corresponding to different % of PAR
(i.e., 75, 54, 36, 10, 1, 0.1%) and filtered [288 mL for Chla and 2.25 L for $a_P(\lambda)$] through 25-
mm Whatman GF/F filters. After filtration, the latter were immediately stored at -80 °C
(liquid N$_2$) in Nunc® cryogenic vials until analysis. Liposoluble pigments were sampled at all
depths (LOV laboratory data, OUTPACE data basis, J. Ras). In addition, samples were taken
in duplicate at surface and Deep Chlorophyll Maximum (DCM) as part of a NASA satellite
validation program. For this, 3 to 4.5 L of seawater was filtered onto 25-mm Whatman GF/F
filters, which were further stored in liquid N$_2$ until analyses at NASA. Watersoluble pigments
(phycoerythrin, PE) concentration were determined for the >10 µm size fractions, therefore
4.5 L of seawater were filtered onto 47-mm Nuclepore filters with pore sizes 10-µm and
stored in Nunc® cryogenic vials. Filters were preserved at -80 °C until analysis at the
laboratory (IRD French Polynesia). For the determination of surface picoplanktonic
population abundances (Bock et al., this issue), water samples were fixed with
paraformaldehyde (final concentration of 0.2%) immediately after sampling, flash frozen in
liquid nitrogen, and stored at -80°C in Nunc® cryogenic vials until analysis. For the





determination of CDOM absorption, 200 mL of seawater were stored in SCHOTT® glass
bottles precombusted (450°C, 6 hours) and rinsed twice with HCL before use, and
immediately filtered on 0.2-µm Micropore filters on Nalgene filtration units rinsed twice with
HCL before each station.
Pump samples were also taken all along three transects in order to increase the frequency of
both pigments and IOPs' surface measurements (Chla, HPLC-NASA) in areas characterized
by important *Trichodesmium* spp. surface slicks : the "Simbada" transect, with 7 pump
samples between SD3 and SD4 in the Melanesian archipelago (MA) (Appendix B), and the
High Frequency HF1 transect (31samples), around LDA, and the HF2 transect in the Fijian
archipelago (FI) near LDB (42 samples) (Dupouy, OUTPACE data basis). Besides
radiometric measurements and water sampling, *in situ* measurements were also conducted for
the determination of *Trichodesmium* spp. colonies and backscattering coefficients (see
below).

**2.4 Phytoplankton abundance**

###### 2.4.1 FTL$_{Trichodesmium}$ abundance: large *Trichodesmium* spp. colonies

An Underwater Vision Profiler 5 (UVP5), serial number Sn003, pixel size ca. 0.147
mm; (Picheral et al., 2010) was deployed fixed to the CTD. The device emitted flashes of red
LED light that illuminates 0.95 L of water. Images of all particles within the illuminated area
were recorded and analyzed for abundances in defined size ranges. Objects larger than 30
pixels were saved and uploaded on ecotaxa (http://ecotaxa.obs-vlfr.fr/prj/37) and further
analyzed by a taxonomist. From 190074 objects recovered, 100342 were identified as "fiber
tricho like *Trichodesmium*" (FTL$_{Trichodesmium}$)", i.e. all particles of *Trichodesmium* with
fusiform-shape (tuff form) and round-shape (puff form) colonies from < 200 µm to 2-5 mm in
size. FTL$_{Trichodesmium}$ is assumed to be mostly *Trichodesmium* colonies with the risk that a
small quantity of fibers is interpreted as diatoms chains. Contrary to a classical counting at the
microscope, no abundance of free filaments is available, although these filaments represent
often a significant part of the *Trichodesmium* assemblage (Carpenter et al., 2004). The
FTL$_{Trichodesmium}$ abundance is calculated in N m$^{-3}$ at 5-m depth intervals (Picheral et al., 2010)
providing FTL$_{Trichodesmium}$ "vertical concentrations" at each cast. Surface FTL$_{Trichodesmium}$
abundance was selected for the surface at each station of the transect. The FTL$_{Trichodesmium}$
abundance at 5m water depth was generally underestimated compared to that at 10-m and 15-
m depths (possibly due to smaller size of colonies). Therefore, the value at 10 m was selected



as representative of the abundance of the surface layer. As different FTL$_{Trichodesmium}$ abundance
profiles were done during the day (from 1 to 5 depending on the station), a daily average at 10
meters of the FTL$_{Trichodesmium}$ abundance was made. Daily average, maximum value of the day,
and the FTL$_{Trichodesmium}$ abundance of the noon profile (i.e. the nearest from the time of the
optical profile) showed no statistical difference. For the long duration stations, an average on
7 days of the 10m-FTL$_{Trichodesmium}$ abundance was calculated as representative of the station.
As an attempt to estimate a trichome concentration, photographies with a Dino-Lite
hand-held Digital Microscope covering the totality of the filtered surface on the GF/F filters
used for absorption measurements were used. Colonies were first visually enumerated. The
uncertainty on this colony visual enumeration was estimated at 20%. For estimating trichome
concentration (L$^{-1}$), a number of 10 trichomes per colony was arbitrarily chosen (representing
an average of each size class and shapes; Dupouy et al., in prep.)

**2.4.2 Picoplankton**
Surface picoplankton population abundances were estimated by flow cytometry using
a BD Influx flow cytometer (BD Biosciences, San Jose, CA, USA). *Prochlorococcus* (Proc),
*Synechochoccus* (Syn) and picoeucaryotes (peuk) were enumerated using the red and orange
fluorescence, while non-pigmented bacteria and protist groups were discriminated in a sample
aliquot stained with SYBR Green I DNA dye, as described in Bock et al (this issue). Using a
forward scatter detector with the "small particle option" and focusing at 488 and 457 nm (200
and 300 mW solid state, respectively) laser into the same pinhole greatly improved the
discrimination between the dim signal from Pro at the surface and background noise in
unstained samples. Nanoeucaryotes were not further differentiated from peuk. Cell
abundances of Proc, Syn, peuk and bacteria showed a vertical and uniform abundance
distribution due to their mixing in the 0-30m layer (Bock et al., this issue).

**2.5 Chlorophyll a, phycoerythrin and pigment analyses**
For Chla determination by the fluorimetric method, filters were extracted with 5 mL
methanol in darkness over a 2 h period at 4 °C and quantified using a Trilogy Turner
fluorometer according to Le Bouteiller et al. (1992) for samples collected over the entire 0-
150 m water column. Surface HPLC pigments (surface, DCM) were measured according to
the NASA protocol and provided monovivyl-Chla (MV-Chla), divinyl-Chla (DV-Chla), and
all accessory pigments, photosynthetic and photoprotective carotenoids (Hooker et al., 2012).





PE was extracted in 50/50 glycerol/phosphate buffer. Quantification of this pigment were
obtained from the area below the fluorescence excitation curve, using a calibrating procedure
previously described (Wyman, 1992; Lantoine and Neveux 1997; Neveux et al., 2006). An
estimate of the relative contribution of each phytoplankton group in terms of Chla was
calculated in Excel from the pigment/Chla ratio found in CHEMTAX (Higgins et al. 2011).
Furthermore, pigment ratios were also used to estimate the relative importance of different
size categories in terms of Chla pico-, nano- and micro-plankton (Ras et al., 2008). The
proportion of Proc to total Chla (TChla) biomass was estimated from the DV-Chla/TChla
ratio. It usually represents a high proportion due of its high abundance despite of its small size
(Grob et al., 2008).

**2.6 *Trichodesmium* concentration algorithms from pigments**
As true microscopic determination of *Trichodesmium* abundance was not realized at
each station during the OUTPACE cruise, we used algorithms to derive trichome
concentrations from pigment concentrations (chlorophylls, zeaxanthin, PE>10µm) and flow
cytometric cell countings. Using a constant PE concentration per trichome (196 pg trichome$^{-1}$)
and a constant Chla per trichome (100 pg cell$^{-1}$) as in Tenorio et al. (in press), calculations of
trichome concentration (L$^{-1}$) could be done both from PE > 10 µm, or Chla > 10 µm,
assuming that other autotrophic organisms have a negligible contribution in this large size
fraction. As fractionated Chla (Chla > 10 µm), however, was not available for OUTPACE,
Total MV-Chla was used, which corresponds to the sum of Chla from Syn and
*Trichodesmium*, and all eukaryotic phytoplankton cells (pico-, nano-, and
microphytoplankton), to estimate MV-Chla from other components of the autotrophic
community and subtract them from the Total MV-Chla. MV-Chla associated with Syn and
peuk was estimated using measured cell concentrations and the Chla per cell values obtained
on cultures grown at high light intensity (Laviale and Neveux, 2011), i.e., 1.2 fg cell$^{-1}$ for Syn
and 10 fg cell$^{-1}$ for peuk (assuming a concentration intermediate between the one of
*Micromonas pusilla* and *Ostreococcus*). The zeaxanthin from *Trichodesmium* spp. was also
estimated from total zeaxanthin using constant sizes for Syn and Peuk Laviale and Neveux
(2011) and pigments/Chla ratios from Carpenter et al. (1993). We compared then estimations
of *Trichodesmium* from these pigment algorithms to FTL$_{Trichodesmium}$ abundance and trichome
concentration estimated from visual counts.



**2.7 Particulate and CDOM absorption and backscattering measurements**

Light absorption spectra were measured directly with filters soaked in filtered sea water, by referencing them to an equally soaked empty filter. Measurements were done in single-beam Beckman DU-600 spectrophotometer. Absorbance (optical density) spectra were acquired between 300 and 800 nm in 2-nm steps. All spectra were shifted to zero in the infrared by subtracting the average optical density between 750 and 800 nm. Finally, optical densities were corrected for the pathlength amplification effect using and then converted into the total particulate absorption coefficients [$a_P(\lambda)$ in m$^{-1}$] (Dupouy et al., 1997; 2003; 2008; 2010). The $a_P$(330 nm) to $a_P$(676 nm) ratio was calculated as an index of photoprotective mycosporine-like amino acids (330 nm: absorption maximum of shinorine) from all phytoplankton species (676 nm, absorption maximum of Chla), as in Ferreira et al. (2013). CDOM absorption spectra were measured on board with a 200-cm pathlength liquid waveguide capillary cell (LWCC, WPI) as described in Martias et al. (2018). Peaks at 350 nm were visible in most of the CDOM spectra, except for LDB and SPG stations (not shown).

Backscattering coefficients were determined as described in Dupouy et al. (2010) from a Hydroscat 6 (HOBILabs, Inc) at 6 wavelengths (412, 442, 510, 550, 620 and 676 nm). Only stations SD1 to SD6 and LDA, days 1-5, were available, with the particulate backscattering obtained by subtracting the backscattering coefficient of pure water (Morel, 1988). Backscattering coefficients of surface oligotrophic waters (SD13, LDC, SD14, SD15) which are supposed to depend deeply from TChla according to Huot et al. (2008) for the South East Pacific, were deduced from Chla using a Look-Up Table of Diapalis data obtained in the Loyalty Channel (Dupouy et al., 2010).

**2.8 Statistics**

Ocean Data View sections Schlitzer, R., Ocean Data View, http://odv.awi.de, 2016 was employed for the spatial representation of biogeochemical parameters over the vertical (0-150m). The spatial interpolation/gridding of data was performed using Data-Interpolating Variational Analysis (DIVA). Principal component analysis (PCA) was conducted on the basis of Pearson's correlation matrices using XLSTAT 2011.2.05 for the surface stations, for AOPs and TChla (HPLC).

## 3 Results

### 3.1 Distributions of $nL_w(\lambda)$, $K_d(\lambda)$ and $Z_{10\%}(\lambda)$

Along the OUTPACE transect, $nL_w(\lambda)$ showed a large range of values and spectral shape (Fig. 2a). In the UV (305–380 nm), violet (412 nm), and blue (443 and 490 nm) ranges, $nL_w(\lambda)$ were the lowest in the Melanesian archipelago's (MA), increasing towards the South Pacific Gyre (SPG) (SD14-SD15, LDC), which exhibited the highest $nL_w(\lambda)$. For all the wavebands, with the exception of the green one (565 nm), $nL_w(\lambda)$ at SD14 and LDC was higher than the 90th percentile, and $nL_w(\lambda)$ at SD9 and LDB were lower than the 10th percentile (Fig. 2b). Values of $nL_w(\lambda)$ in this violet-blue domain were similar than those measured in the most oligotrophic oceanic areas at the Eastern part of the OUTPACE transect (Tedetti et al., 2010). For example, in the center of the SPG (15°S-30°S, 126°W-86°W 160°W), $nL_w(412)$, $nL_w(443)$, and $nL_w(490)$ reached up to 4.5, 4, and 2 µW cm$^{-2}$ sr$^{-1}$ nm$^{-1}$, respectively for TChla concentrations < 0.022 mg m$^{-3}$ and a DCM at 180 m. LDB had a characteristic spectrum with waters greener than all other stations (Fig. 2a). The low $nL_w$ at LDB corresponds to a surface TChla accumulation of 1 mg m$^{-3}$ on a surface physical front (Rousselet et al., this issue). This dark green color was astonishing from the ship deck while profiling of the Satlantic instrument. Moreover, the GF/F filters used for absorption showed an orange-yellow color when observed under the Dino-Lite microscope. Such color was not observed in the MA, and is typical of small picoplanktonic cells as Pro and Syn.

Table 1 displays $K_d(\lambda)$ values at the four UV wavelengths and for the whole PAR domain. For all stations, $K_d(\lambda)$ decreased from the UV-B to UV-A spectral domain (Table 1). From the MA to the FI, $K_d(325)$ was high from SD1 to SD6, then decreased from SD7 to SD12, and showed a peak at LDB, and minimum at the SPG stations. During the long duration stations, $K_d(325)$ variations (not shown) reflected those of TChla with values decreasing from day 1 to 5 at LDA (0.18 to 0.15  m$^{-1}$), at LDB (0.22 to 0.17 m$^{-1}$ ). They stayed stable at LDC. $K_d(PAR)$ (Table 1) showed the same distribution within the range 0.016 m$^{-1}$ (LDC and SD14) to 0.028 m$^{-1}$ (LDB). Typical values of $K_d(PAR)$ in oligotrophic waters associated to a deep DCM of 165 m and a TChla of 0.037 mg m$^{-3}$ were measured at SD13-SD14-SD15-LDC. For comparison, such values are close to that found in South East Pacific during BIOSOPE cruise (08–35°S, 142–73°W) (Tedetti et al., 2007) and much lower than that reported for the oligotrophic water of NW Mediterranean Sea (Sempéré et al., 2015).





Maxima of $Z_{10\%}(380)$ (Table 1, Fig. 3) were found in the FI at LDC and SD15 (100-
120 m, for a TChla of 0.02 mg m$^{-3}$) and was comparable to those reported for the clearest
natural waters in SPG (Tedetti et al., 2007). Conversely, stations exhibiting the lowest $Z_{10\%}$
(SD1, 40 m) were found in the MA and at the LDB frontal station in the FI (DCM of 41 m,
TChla = 0.433 mg m$^{-3}$). The 1$^{st}$ optical depth determined in the UV-visible varied from 13
(LDB-Day3) to 28 m (SD14).

**3.2 Pigment composition and abundance of phytoplanktonic groups**

**3.2.1 FTL$_{Trichodesmium}$ abundance derived from Underwater Vision Profiler**
The UVP5 FTL$_{Trichodesmium}$ abundance showed a wide range of values along the
transect SD1-SD15 (Fig. 4a, Table 1) in the SWTP. It was essentially concentrated in the
upper 30 m although some were still visible below 30 m. The maximum was obtained at SD1
(4000 N m$^{-3}$) and rapidly dropped to 2000 N m$^{-3}$ at SD2 to stabilize between 200 and 500 N
m$^{-3}$ at the east of SD4. It progressively decreased from West to East. Still visible at SD5
(170°E), it vanished at SD7, where the maximum of FTL$_{Trichodesmium}$ abundance was located
deeper and finally disappeared between SD8 and SD11. A second maximum of
FTL$_{Trichodesmium}$ abundance was found at LDB at 50 m with an exceptional value of 3500 N m$^{-3}$
at Day1. The continuity of FTL$_{Trichodesmium}$ abundance was described for the first time over the
total water column in the SWTP. FTL$_{Trichodesmium}$ abundance allowed one to classify 3 groups
of stations, according to its Log10 of N m$^{-3}$. The 1$^{st}$ group was composed by the stations SD1
to SD7, SD8 (but not SD9) and included both LDA in the western MA and LDB in the FI
(log10 >2.7). The 2$^{nd}$ group was composed by SD8 to SD12 with medium concentrations
(2<log10 < 2.7). Finally, the 3$^{rd}$ group contained the stations SD13, SD14, LDC and SD15
characterized by no or very low FTL$_{Trichodesmium}$ abundance (log10 < 1).
**3.2.2 Picoplankton abundance and influence on TChla biomass**
Picoplankton predominance was typical of oligotrophic waters (Neveux et al., 1999;
Buitenhuis et al., 2012; Bock et al., this issue). The Syn abundance was particularly high in
the surface layer in the MA at SD3-LDA (> 22 10$^3$ cells mL$^{-1}$) until the intermediate area of
the Fijian basin (FI), except the surface maximum at LDB (> 100 10$^3$ cells.mL$^{-1}$). The Proc
abundance peaked at LDB with more than 9.10$^5$ cells mL$^{-1}$ in the upper surface layer and the
Peuk was high at the DCM only.
**3.2.3 Chla, PE and accessory pigments**
HPLC pigment analyses revealed the occurrence of three major pigments identified as





diverse Chla, zeaxanthin and β-carotene, classically observed in marine cyanobacteria
(Higgins et al., 2011). Pigment concentrations from LOV were used as they are available for
each station and depth at Fig. 4a-c (HPLC LOV laboratory data, OUTPACE data basis, J.
Ras). The 0-150 m section of zeaxanthin, the main photoprotectant carotenoid contained in all
cyanobacteria (Syn, Proc + *Trichodesmium*), showed an extremely rich surface layer (> 0.15
mg m$^{-3}$) from 0 to 50 m and almost continuously from SDA to SD12. A strong maximum was
observed at the frontal LDB (Fig. 4b). TChla section (Fig. 4c) showed high values in the MA
near the islands of New Caledonia-Vanuatu (SD1 to SD6) (with a maximum of 0.352 mg m$^{-3}$
at SD1 at 5 m), and a DCM oscillating between 70 and 110 m (Table 1), with a TChla (0.534
mg m$^{-3}$) and an extremely shallow DCM (52 m) at the frontal LDB.  Surface PE > 10 µm
values (indicative of *Trichodesmium*) showed two spots of high concentrations (Fig. 4e). The
first spot is located in the Western part of the MA (SD1 to SD5), and the second is located at
LDB. PE was low in the central part of the transect (between SD6 and SD12), and was near 0
in the SPG. Higher surface values of TChla and PE>10 at LDA and LDB (Fig. 4d, e) are from
pump samples, and provided higher values than surface Niskin samples.

DV-Chla of Proc (Fig. 5a) increased from West to East and showed a prominent

maximum of 0.18 mg m$^{-3}$ from 0 to 30 m at the frontal LDB with proportions, of 22% in the
MA, 39% in the FI, and up to 39% in the SPG (and 45% at LDB). The decomposition of MV-
Chla (paragraph 2.6) showed (Fig. 5a) that Syn+Peuk were not important contributors to MV-
Chla biomass (< 0.011 mg.m$^{-3}$ on average) nor the sum of nano- + micro-plankton (< 0.028
mg m$^{-3}$). Tricho-Chla was then between 0.15 mg m$^{-3}$ in the MA, 0.03 mg m$^{-3}$ in the FI, with a
high value of 0.08 mg m$^{-3}$ at LDB and < 0.02 mg m$^{-3}$ in the SPG. Its contribution to TChla
(Fig. 5b) varied from 52 to 33% between MA and FI, and still 23% of TChla in the SPG
(SD12-LDC). Its % contribution at LDB was lower because of a high DV-Chla. Note that
identical contributions were calculated either using LOV or NASA surface pigments (TChla-
LOV and TChla-NASA was highly correlated (TChla$_{LOV}$ = 0.86 × TChla $_{NASA}$; r$^2$ = 0.93, p <
0.05, n =15), and zea$_{LOV}$ = 0.70 × zea$_{NASA}$; r$^2$ = 0.78, p < 0.05, n =15). The contribution of
*Trichodesmium* to zeaxanthin followed roughly the same pattern, except at SD1 and was
somewhat higher between SD8-SD11.

### 3.2.4 *Trichodesmium* abundance

The trichome concentration (L$^{-1}$) estimated from PE > 10 µm (paragraph 2.6)  ranged

from 0 in LDC SD14-15 to 4580 trich L$^{-1}$ at SD1 (Fig. 6). The one estimated from Chla (or
Chla.Trich) ranged (Fig. 6) between 3692 trich L$^{-1}$ at SD1, 144 at SD13 and 1379 trich L$^{-1}$ at





LDB. The difference between the estimation from PE >10µm or Chla at SD1 may be due to
patchiness leading to a high variability of colony abundance in water samples (Fig. 6).
Trichome concentration estimates from pigments showed the same pattern that the one
obtained from visual counts (Fig. 6).
Fig. 7a shows significant regressions between trichome concentrations estimated from
PE > 10 µm or Chla.Trich and FTL$_{Trichodesmium}$ abundance. The relatively high slopes of the
correlation (900 and 600 as the factor between colonies and trichomes, from PE and Chla
respectively) indicate aggregation processes. The correlation (Fig. 7b) between Chla-Tri and
our visual counts ($r^2$=0.80) was also significant.

### 3.3 Absorption and backscattering coefficients, photoprotection index

*Trichodesmium*-rich backscattering coefficient (bb-H6) was higher by a factor of 2 at
the stations with the highest *Trichodesmium* concentrations (SD1 and SD2) compared to those
with lower *Trichodesmium* concentrations (SD2 to SD6) and an oceanic station off New
Caledonia (Fig. 8a). It showed large troughs due to absorption maxima at these wavelengths
in the blue channel (Fig. 6a-d). The section from 0 to 150 m of b$_b$-H6 showed that the high
backscattering layer characterizes the 0-10 m in the MA (no data were collected after SD5).
Typical spectra of particulate absorption for *Trichodesmium*-rich waters (SD1, SD2,
and other stations of the highest FTL$_{Trichodesmium}$ abundance group) exhibit the 2 MAAs
absorption peaks at 330 and 360 nm (Fig. 9a). These peaks are visible on *in vivo* spectra
(Dupouy et al., 2008) and their amplitude though enhanced by freezing (Laurion et al., 2009)
are used in many studies to show the degree of photoprotection by MAAs against UV
(Ferreira et al., 2013). These peaks never appear at the surface in low *Trichodesmium*
concentrations (the medium FTL$_{Trichodesmium}$ abundance group; Fig 9b). Sections from 0 to 150
m of a$_P$(330) and a$_P$(440) (Fig. 9c, upper and lower panels) exhibit the impact of MAAs in the
upper layer at 350 nm, ant the effect of the DCM at 442 nm. High values (> 80) of a$_P$(330)
(0.4 m$^{-1}$) and of the a$_P$(330)/a$_P$(676) ratio were measured from 0 to 25 m, and abruptly felt to
20 below 30-m depth (Fig. 10a). A reasonable relationship was found between UVP-5
FTL$_{Trichodesmium}$ abundance and a$_P$(330) when considering all depths (from 0 to 150 m)
(FTL$_{Trichodesmium}$ abundance = 0.43 a$_p$(330) -2.1, $r^2$ = 0.57, n = 100)  (Fig. 10b). At the surface
(Fig 10c), the MAAs index was variable along the transect, and was not tightly related to
*Trichodesmium*. Discrepancies are seen in some stations such as SD6 and SD10 (with high
values of the MAAs index and low *Trichodesmium* abundance), as at these stations,





phytoplanktonic cells other than *Trichodesmium* might also be protected. Indeed, MAA
pigments are produced by all phytoplankton groups (Carreto and Carignan, 2011) when
exposed to high $nL_w(UV)$ values. Their MAA's index might also be influenced by the value
of $a_P(676)$ because of different package effect at this wavelength linked to the size. MAA's of
other groups show generally only one peak at 320 nm as in the South Eastern Pacific at
BIOSOPE (Bricaud et al., 2010) or at 330 nm (large phytoplankton in the Argentina
continental shelf; Ferreira et al., 2003). These other groups were in low abundance at the
surface at OUTPACE (as shown by the size index from HPLC).

### 3.4 Relationships between AOPs and pigments


In the present study, Chla was well correlated to all $nL_w(\lambda)$ ratios $[nL_w(\lambda)/nL_w(565)]$
with $r^2$ varying from 0.79 to 0.83 (Fig. 11). The relationships between and Chla showed the
same fits as for BIOSOPE (except at 305 and 325 nm). These good relationships obtained
even in the UV domain (where Chla does not absorb) were already observed in the South East
Pacific, for equivalent ranges, and attributed to the fact that CDM substances covary with
Chla (Tedetti et al., 2010).

### 3.5 Potential influence of *Trichodesmium* on the distribution of $nL_w(\lambda)$


To better assess the influence of *Trichodesmium* on the distribution of $nL_w(\lambda)$ values,
the 8 radiances of the South West Pacific OUTPACE cruise (this study) and of the South East
Pacific data (BIOSOPE cruise, 2004) were statistically analyzed. Fig. 12a-d represents the
results of a principal component analysis (PCA) operated separately on $nL_w(\lambda)$ values and
TChla concentrations for the two cruises. In the South West Pacific, the two principal
components (PCs) represent 93% of total variance (Fig. 12b). The graph of correlations
between PCs and the variables (Fig. 12a) indicates that UV and visible $nL_w(\lambda)$ are distributed
along the PC1 axis, with all radiances on the right side, except 565 nm. This 1st axis (83% of
total variance) indicates an effect of Chla on $nL_w(\lambda)$, with all $nL_w(\lambda)$ being higher at low Chla
(blue waters) and lower at high Chla (mesotrophic waters), except at 565 nm, where on the
contrary $nL_w$ increases with Chla. Oligotrophic stations are on the right side and mesotrophic
stations on the left. PC2 represents 9.4% of the total variance. The variables that have
significant correlation with PC2 are $nL_w(565)$, (Chla rich waters) and $nL_w(490)$ (Chla poor
waters), both on the upper side of the PC2 axis. A series of stations is positively linked to this
PC2 axis (LDB4, SD1, SD2, LDA-2, SD7) while LDA-3 and LDA-4 are negatively linked to



PC2. The relatively high correlation between PC2 and $nL_w(565)$, minimally influenced by
Chla, suggests that other parameters than abundance (e.g., size, type) might affect $nL_w(565)$ at
the stations with sizeable PC2 values.

In comparison, the first 2 PCs for the South East Pacific dataset represent 95% of the

total variance, with 89% for PC1 and only 5% for PC2 (Fig. 12c). The main difference is that
$nL_w(565)$ is no more linked with PC2 but only to PC1, and that for PC2 $nL_w(490)$ has an
opposite behavior compared to that in the South West Pacific (correlation is negative instead
of positive). At 490 nm, Chla appears to explain most of the $nL_w$ variability. This could reflect
the absence of *Trichodesmium* in the Eastern Pacific. Except for a few stations, the PC2
contribution is much lower, i.e., variability is mostly described by PC1.

## 474    4. Discussion


### 476    4.1 Determination of the contribution of other phytoplankton and filamentous
### 477    cyanobacteria to absorption and backscattering coefficients

The determination of *Trichodesmium's* influence on IOPs compared to other

microorganisms and non-living particles in the sea is a main challenge. Indeed, previous
models showed that absorption is governed by size and intracellular content (Bricaud et al.
1995; 2004; 2010) while backscattering is rather influenced by small particles (< 0.5 µm) of
mineral origin, bubbles and colloids than by soft marine living particles (Loisel et al., 2007;
Stramski et al., 2008). In oligotrophic waters of the South East Pacific, backscattering was
well related to Chla (Huot et al., 2008), and recent studies in the open ocean indicate a strong
correlation with particles (Dall'Olmo et al., 20109; Brewin et al., 2012; Martinez Vincente et
al., 2013; Slade and Boss, 2017). Our H6-backscaterring data at OUTPACE compared to the
ones of Diapalis (not shown) show that backscattering is enhanced in the presence of
*Trichodesmium*. The layer of the highest backscattering coefficient is situated above the 10m-
$FTL_{Trichodesmium}$ and the relationship between the vertical distribution of $b_{bp}$, and the vertical
structure of colonies, detritus and organisms must be explored further. There is a strong link
between particulate backscattering and particulate organic carbon (Stramski et al., 2008;
Evers-King et al., 2017). The organic carbon content of *Trichodesmium* filaments was not
estimated in the South West Pacific as trichomes counts are not yet available at all depths and
stations (Dupouy et al., in prep.). However, we found total algal carbon portions were in the
range 10-50% with a maximum of 75% during the bloom in the Loyalty Channel (Tenorio et





al., in press). Additional work is needed to model influence of *Trichodesmium* in terms of
pigment biomass, and carbon biomass on $nL_w(\lambda)$ values.

**4.2 Specificity of fluorescence and pigmentation of *Trichodesmium* for interpreting**


**satellite Chla imagery**


Is has been shown that in the upper layer of the 0-150m section, particularly in the
western part of the MA, that highest *Trichodesmium* abundance and $a_P(330)$ are well
correlated. At the opposite, $a_P(440)$ is lower than expected for this *Trichodesmium* abundance.
*Trichodesmium*-specific Chla values retrieved from satellite observations, are expected to be 4
times lower because of a shadow effect on absorption of light by colony (until a factor of 4
(Subramanian et al., 1999; McKinna, 2015). It was also noted that the CTD fluorescence
signal was also weak as already noted in the region (Diapalis; Tenorio et al., in press). Last, it
can be attributed to a "deficient" response of large colonies to the laser compared with the
numerous small picoplanktonic cells (Neveux et al., 2010). This can be attributed to a
"deficient" response of large colonies to the laser compared with the numerous small
picoplanktonic cells (Neveux et al., 2010). Note that Chla should be measured also on a
sufficient volume to catch colonies, adjusted as a to expected abundance.

**4.3 The influence of *Trichodesmium*-CDOM to ocean color**


High CDM amount is expected to be associated *Trichodesmium*, either from CDOM
issued from degradation of colonies, and/or from MAAs absorption (Subramaniam et al.,
1999a; Steinberg et al., 2004; Dupouy et al., 2008). MAAs identified by their strong UV
absorption at 332 and 362 nm are the water-soluble pigments asterina-330 and shinorine as
the most abundant, and the mycosporine-like amino acids, like glycine and porphyra-334, and
palythene-360 as minor components. In order to define the best photoprotection index for
*Trichodesmium*, it would be useful to take into account the double absorption peak at 330 and
360 nm and variability of absorption peak at 676nm as a function of size (Bricaud et al.,
2010). Indeed, a complete analysis of the different components of CDM measured during the
cruise over the whole water column has still to be achieved.

**4.4. Contribution of *Trichodesmium* spp. to TChla and ocean color**


All *Trichodesmium* abundance data, obtained from UVP5, pigments and flow
cytometry data, or from visual counts showed a high abundance in the MA that strongly



influences the Chla biomass in the western part of the Melanesian archipelago and a lower abundance in the SPG, with a mean value at LDB. Trichome abundance estimated from the decomposition of pigments were equivalent to the ones enumerated with the microscopy in the region (at 167°E, 21°S, Diapalis data; Shiosaki et al., 2014). The UVP5 counted the largest colonies of the *Trichodesmium* population, i.e. the upper part of the colony size distribution. The factor between these counts and estimated trichome concentrations (1000) depends on the number of isolated or small colonies (unknown) and from other aggregation processes which remain to analyse further. From literature data, this number varies between 200 for the highest to 50 for the lowest (Davis and McGillicudy, 2006; Guidi et al., 2012; Olson et al. 2015) and the largest numbers found here may indicate different proportions in colonies and trichomes in the South Western tropical Pacific. High colony abundance was detected from 0 to 30 meters with the UVP5 even though colonies are detected deeper with this instrument. Their abundance was low. In the region, trichomes are generally found from 0 to 60 meters (Tenorio et al., in press). The high contribution of *Trichodesmium* detected during OUTPACE in the Western part of the Melanesian archipelago (around New Caledonia and Vanuatu) between 158 and 174°E and at the frontal accumulation at 170°W match the large amount of surface mats detected from the satellite in this part of the transect (Dupouy et al., 1988; 2000; 2011) and was observed during the OUTPACE cruise thanks to a new algorithm developed for the region (Rousset et al., this issue). It is the first time that this continuity of *Trichodesmium* is measured with a profiler from 0 to 150 meters on the whole Southwest tropical Pacific. Even if *Trichodesmium* abundance is lower around 180, there might be enough colonies below the surface (less visible by the satellite) to produce mats, as soon as environmental conditions are favourable (as there are observed there, but more episodically) than at 170°E where they are frequent.

Proc was the other dominant group impacting the Chla biomass. Two parts of the SWTP ocean at 174°E (SD7): (1) the western part of the MA between New Caledonia and Vanuatu, impacted by a large contribution by *Trichodesmium* and (2) the eastern part of the transect (FI), more oligotrophic and impacted by *Prochlorococcus* and other picoplanktonic groups. LDB was the only exception showing a high abundance of both *Trichodesmium* and other groups, with TChla proportions of *Trichodesmium*, Syn+Peuk, Micro, Nanoeuk, and Proc of 25, 7, 1.4, 5 and 45 %, respectively.

OUTPACE and BIOSOPE radiometric data show that the South West and South East Pacific surface waters exhibited similar ranges of values for $nL_w(\lambda)$ and Chla (0-0.58 and



0.02-1.3 mg m$^{-3}$, respectively). It should be noticed that this "extreme" value of 1.3 mg m$^{-3}$
was recorded in the Peru upwelling. OUTPACE and BIOSOPE data differed in only two
spectral bands. OUTPACE and BIOSOPE radiance data with the OCR UV-Vis radiometer
differed in only two spectral bands. The PC1 axis was linked to Chla concentration for both
cruises. PC2 was linked to another optically active variable, independent of Chla. For
OUTPACE, the different behaviors in $nL_w(565)$ (yellow) and $nL_w(490$ nm) (green) are
significant compared to the sensitivity of the instrument. The significance of PC2 (linked to
an increase of $nL_w(565)$ for Chla rich waters) is clear. It is absent in the South East Pacific
meaning that all these processes were not occurring during the BIOSOPE cruise, i.e. there is
no effect of particles or PE at high Chla concentrations. Indeed, Huot et al. (2008) showed
that backscattering measured during the BIOSOPE stations (between 41°W and 173°W) is
totally linked to Chla.
The relationship of $nL_w(490)$ to PC2 is more difficult to interpret due to its opposite
behavior between the South West and the South East Pacific. PCA shows that the variability
in $nL_w(490$, green) and $nL_w(565$, yellow) is not totally determined by Chla, as it exists a non-
negligible correlation between PC2 and these green and yellow radiances. However this effect
is reversed between these two areas. One explanation would be that in the presence of
*Trichodesmium*, it is expected a higher backscattering (linked to another factor than Chla) and
a PE fluorescence in the yellow, but also a high backscattering in the green. However, in the
green, there is a possible effect of the absorption by zeaxanthin (the major photoprotecting
pigment, not totally correlated with Chla as shown by the PCA). PCA applied to the only nLw
and Chla variables does not allow one to explain the correlation between PC2 and $nL_w(490)$
(due to a different pigment, PE fluorescence or backscattering). At BIOSOPE, where
$nL_w(490)$ is essentially function of Chla, the zeaxanthin effect would be negligible or totally
linked with the one of Chla.
The spectrum obtained from an optical model of a *Trichodesmium's* bloom at 0.5 mg
Chla m$^{-3}$ showed a similar shape than those from other phytoplankton, but with higher
magnitudes for $nL_w(490)$, $nL_w(510)$ and $nL_w(555)$  (Subramaniam et al., 2002). At 0.5 mg
Chla m$^{-3}$, the magnitude of $nL_w(510)$ was greater than $nL_w(443)$, and for middle
concentrations, from 0.5 and 1.5 mg Chla m$^{-3}$, the model predicted that a peak should be
observed at $nL_w(490)$. At concentrations approximating 2 mg Chla m$^{-3}$ and higher, $nL_w(555)$
exceeded $nL_w(490)$. These two wavelengths were those chosen by Westberry and Siegel
(2006) to set an algorithm to map *Trichodesmium* globally with SeaWiFS. Nevertheless, the



results were not satisfactory around SWTP islands, even in summer when blooms are
numerous (Dupouy et al., 2011). The reason why their algorithm was not successful around
New Caledonia might be due to an inappropriate radiance model at 490nm and 565 nm in the
case of moderate *Trichodesmium* abundance.

**5 Conclusions**
The OUTPACE cruise in the South West Tropical Pacific from 158°E to 160°W
provided a unique set of simultaneous measurements of $nL_w(\lambda)$ in the ultraviolet and visible
domains, pigments, and *Trichodesmium* and picoplanktonic cell abundance along the whole
transect during a summer bloom. *Trichodesmium* abundance given by the UVP5 (i.e., largest
colonies) was well correlated with the trichome concentration estimated at the surface from
pigment algorithms and visual counts. The factor of 600-900 observed between large colonies
and trichome concentration is indicative of aggregation processes, and is also specific to all
cameras towered or lowered in the ocean. *Trichodesmium* abundance was also well correlated
with the absorption peak of MAA's, i.e. $a_P(330)$ and the photoprotection index. This
demonstrates that UVP5 is a well adapted instrument for exploring the variability of
*Trichodesmium* in the water column, and that $a_P(330)$ or photoprotection index, is a useful
parameter to quantify the latter. The weak CTD-fluorescence and blue absorption observed in
rich-*Trichodesmium* waters tend to underestimate *Trichodesmium* abundance if used on
profilers while the backscattering (high coefficient, spectral troughs) tend to correctly
estimate *in situ* aggregations. Along the 165°E-170°W transect, *Trichodesmium* together with
*Prochlorococcus* represented the major part of TChla (the other groups were negligible).
*Trichodesmium* contribution to TChla was the highest in the Western part of the Melanesian
Archipelago (around New Caledonia and Vanuatu), regularly decreased to the East, in the
vicinity of the Fiji Islands, reaching a minimum in the South Pacific gyre stations where the
*Prochlorococcus* contribution to TChla was higher. It is the first time that this continuity from
0 to 150 m of *Trichodesmium* abundance is measured with a profiler on the whole South West
tropical Pacific. Then, even if *Trichodesmium* abundance is lower south of Fijian Islands,
there might be enough colonies below the surface (less visible by the satellite) to produce
mats, as soon as the environment is favourable (as it is observed, but more episodically than at
170°E).
During OUTPACE, the relationship between normalized water radiance, $nL_w$, and
Chla was generally similar to that found in the Eastern Tropical Pacific during BIOSOPE. In



particular, radiance ratios were related to Tchla in the visible and the UV domain interpreted
as a strong coupling between the UV-absorbing Chromophoric Dissolved Material and Chla.
Principal component analysis (PCA) of OUTPACE data showed that $nL_w$ in the ultraviolet
and visible were strongly correlated to Chla except in the green and yellow (490 and 565
nm). These results, as well as differences in the PCA of BIOSOPE data, suggested that
that $nL_w$ variability in the green and yellow radiance during OUTPACE was influenced by
other variables associated with *Trichodesmium* presence, namely a specific backscattering
coefficient, phycoerythrin fluorescence, and/or zeaxanthin absorption. These green (490 nm)
and yellow (565 nm) wavelengths are often chosen in *Trichodesmium* detection algorithms.
Indeed, more work is required to explain the PCA results. It would be useful to include
backscattering coefficient, PE, photoprotecting carotenoids from HPLC and *Trichodesmium*
abundance at all depths into the PCA analysis. Also, it would be useful to compare our
measured radiance in this *Trichodesmium* bloom to modeled radiance of classical
phytoplankton to highlight potential anomalies, and last, to decompose the effect of
*Trichodesmium* specific IOPs and pigments on radiance. While detecting *Trichodesmium* mats
with the "red edge" is essential (Rousset et al., this issue), as this part of colonies may also
actively fix N2, exploring the green-yellow change in ocean color detected here at regular
*Trichodesmium* concentrations is probably the only way to assess true nitrogen fixation rates
in the SWTP.
*Acknowledgments.* The authors thank the crew of the R/V L'Atalante from operation at sea.
We acknowledge Joséphine Ras, CNRS, for the OUTPACE HPLC databasis and Crystal
Thomas, NASA, for surface and DCM HPLC data. We thank Mireille Pujo-Pay for her
scientific advices on board R/V *Atalante*. We thank Benjamin Blanc for sorting and validating
images from the UVP-5. We acknowledge David Varillon, IRD US IMAGO, and Philippe
Gérard for Chla analyses and the administrative staff of Center of Noumea. Jacques Neveux
and Rüdiger Röttgers are acknowledged for helpful discussions during the elaboration of the
manuscript. This is a contribution of the OUTPACE (Oligotrophy from Ultra-oligoTrophy
PACific Experiment) project (https://OUTPACE.mio.univ-amu.fr/) funded by the French
research national agency (ANR-14-CE01-0007-01), the LEFECyBER program (CNRS-
INSU), the GOPS program (IRD) and the Centre National d'Etudes Spatiales (BC T23, ZBC
4500048836). The OUTPACE cruise (http://dx.doi.org/10.17600/15000900) was managed by
MIO Institute from Marseilles (France). The National Science Foundation supported SD
under grant OCE-1434916.



**Table 1.** Main characteristics of the OUTPACE stations.

| Station | Longitude | Latitude | Date | UT time | TChla (mg m$^{-3}$) | FTL$_{Tric}$ _hodesmium_ (N.m$^{-3}$) | DCM (m) | PE (mg m$^{-3}$) | $K_d(\lambda)$ (m$^{-1}$) 305 nm | 325 nm | 340 nm | 380 nm | PAR |
|---|---|---|---|---|---|---|---|---|---|---|---|---|---|
| SD1 | 159°54' E | 18°00' S | 21 Fev.15 | 20h00 | 0.352 | 4125 | 101 | 1.15 | 0.173 | 0.116 | 0.093 | 0.05 | nd |
| SD2 | 162°07' E | 18°37' S | 22 Fev. 15 | 21h45 | 0.278 | 2430 | 70 | 0.122 | 0.194 | 0.119 | 0.099 | 0.057 | 0.026 |
| SD3 | 164°54' E | 19°00' S | 24 Fev.15 | 03h45 | 0.236 | 445 | 70 | 0.08 | nd | nd | nd | nd | 0.028 |
| LDA* | 164°41' E | 19°13' S | 25 Fev. 15 | 13h00 | 0.220 | 974 | 100 | 0.10 | 0.074 | 0.041 | 0.029 | 0.012 | 0.024 |
| SD4 | 168°00' E | 20°00' S | 04 Mar. 15 | 08h30 | 0.199 | 1674 | 70 | 0.43 | nd | nd | nd | nd | nd |
| SD5 | 170°00'S | 22°00' S | 05 Mar. 15 | 05h45 | 0.258 | 902 | 70 | 0.26 | nd | 0.124 | 0.083 | 0.048 | nd |
| SD6 | 172°08' E | 21°22' S | 06 Mar. 15 | 03h15 | 0.265 | 935 | 130 | 0.05 | 0.159 | 0.108 | 0.087 | 0.044 | 0.025 |
| SD7 | 174°16' E | 20°44' S | 07 Mar. 15 | 00h00 | 0.186 | 1059 | 110 | 0.08 | 0.117 | 0.073 | 0.053 | 0.009 | 0.019 |
| SD8 | 176°24' E | 20°06' S | 07 Mar. 15 | 21h00 | 0.138 | 165 | 120 | 0.03 | 0.143 | 0.087 | 0.065 | 0.026 | 0.021 |
| SD9 | 178°39' E | 20°57' S | 08 Mar. 15 | 22h15 | 0.236 | 569 | 120 | 0.08 | 0.152 | 0.097 | 0.074 | 0.041 | 0.020 |
| SD10 | 178°31' W | 20°28' S | 10 Mar. 15 | 00h00 | 0.113 | 127 | 120 | 0.04 | 0.139 | 0.086 | 0.065 | 0.034 | 0.020 |
| SD11 | 175°40' W | 19°59' S | 10 Mar. 15 | 21h45 | 0.185 | 188 | 110 | 0.09 | 0.137 | 0.082 | 0.06 | 0.024 | 0.033 |
| SD12 | 172°50' W | 19°29' S | 11 Mar. 15 | 21h00 | 0.133 | 139 | 120 | 0.04 | 0.116 | 0.069 | 0.051 | 0.027 | 0.020 |
| LDB* | 170°52' W | 18°14' S | 15 Mar. 15 | 23h00 | 0.433 | 2950 | 52 | 0.24 | 0.172 | 0.11 | 0.087 | 0.054 | 0.028 |
| SD13 | 169°04' W | 18°12' S | 21 Mar. 15 | 22h30 | 0.0357 | 4 | 125 | 0.00 | nd | nd | nd | nd | nd |
| LDC* | 165°45' W | 18°41' S | 23 Mar. 15 | 01h00 | 0.0231 | 0.82 | 135 | 0.01 | 0.189 | 0.116 | 0.09 | 0.054 | 0.020 |



| | | | | | | | | | | | | |
|---|---|---|---|---|---|---|---|---|---|---|---|---|
| SD14 | 163°00' W | 18°25' S | 30 Mar. 15 | 01h30 | 0.045 | 0 | 165 | 0.04 | nd | 0.056 | 0.04 | 0.023 | 0.018 |
| SD15 | 160°00' W | 18°16' S | 31 Mar. 15 | 00h00 | 0.061 | 0 | 110 | 0.00 | 0.097 | 0.054 | 0.039 | 0.021 | 0.016 |

TChla: average concentrations in total chlorophyll a (monovinyl Chla + divinyl Chla) in surface waters derived from HPLC

analyses, based on duplicate analyses (CV < 8%). FTL$_{Trichodesmium}$ abundance: determined using underwater vision profiler 5 (UVP5).

DCM: deep chlorophyll maximum. PE: phycoerythrin. K$_d$($\lambda$): diffuse attenuation coefficient for downward irradiance in the UV (305, 325, 340,

380 nm) and PAR (400-700 nm) domains.

* Values for Long Duration stations, i.e., LDA, LDB and LDC, averaged over 7 days.




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

**FIGURE LEGENDS**
**Figure 1.** Chlorophyll composite from MODIS on the period of the OUTPACE cruise. The
positions of the short (long) duration stations are shown by cross (plus) symbols. The ocean
color satellite products are produced by CLS. New Caledonia, Vanuatu and Fiji islands at
165°, 170°E, and 180°E. Tonga Trench at 170°W (190°E).

**Figure 2.** OUTPACE AOPs: a) Box-and-whisker plots for the distribution of $nL_w(\lambda)$ in the
UV (305, 325, 340, and 380 nm) and visible (412, 443, 490, and 565 nm) spectral domains
determined between $0^-$ and 30 m in the Western tropical South Pacific at stations in the
Melanesian arch. (MA, SD1-SD7 and LDA), Fijian arch. (FI, SD8-SD11), and South Pacific
Gyre (SPG, SD13, LDC, SD14, SD15). The outliers stations are indicated on the upper left
(see text). b) $nL_w(\lambda)$ *versus* wavelength with a color-code depending on TChla (in red: high
concentrations, in black: median concentrations, in blue oligotrophic).

**Figure 3**. OUTPACE AOPs (continued). $Z_{10\%}(\lambda)$ at 305 nm (UV-B), and 325, 340 and 380
nm (UVA-A) at all stations during OUTPACE in the Western tropical South Pacific with a





911 color-code depending on TChla (in red: high concentrations SD1 to SD7, Melanesian
912 archipelago), in black: median TChla: medium concentrations, SD8 to SD11 around Fiji
913 Islands, in blue low concentrations SD12 to SD15 including LDC (Table 1) with the frontal
914 station LDB in green.

915

916 **Figure 4.** Sections from 0 to 150 m of a) Abundance of Fiber Tricho Like$_{Trichodesmium}$ (N.m$^{-3}$),
917 b) Zeaxanthin concentration (mg.m$^{-3}$), c) TChla concentration (mg.m$^{-3}$) by HPLC-LOV (J.
918 Ras) d) Surface maps of TChla HPLC-NASA (mg.m$^{-3}$) and e) PE > 10 µm by
919 spectrofluorimetry (mg.m$^{-3}$). Short transects data from pump samples (5 m depth) at 165°E
920 and at 170°W are included. Ocean Data View sections Schlitzer, R., Ocean Data View,
921 http://odv.awi.de, 2016. Station numbers along the transect are indicated.

922

923 **Figure 5.** Surface values along the transect of a) DV-Chla (mg m$^{-3}$) and different fractions of
924 MV-Chla (mg m$^{-3}$) using HPLC and flow cytometry (Chla-nano+micro), Chla-Syn+Peuk)
925 allowing to extract Chla from *Trichodesmium* (Chla-Trich.), b) Zeaxanthin (mg m$^{-3}$) (left
926 axis) and % of TChla and % zeaxanthin by *Trichodesmium* (right axis). All pigments from
927 HPLC NASA. Station numbers along the transect are indicated on the X axis. Main
928 longitudes (E, W) are indicated above.

929

930 **Figure 6.** Surface values along the transect of the trichome concentration from visual counts
931 and estimated from the Chla, zeaxanthin and PE of *Trichodesmium* (Trich.(Chla), Trich(zea),
932 Trich.(PE>10µm)) (N L$^{-1}$) (left axis) and of FTL$_{Trichodesmium}$ abundance (colony counts by
933 UVP5) at 10 m (N L$^{-1}$) (right axis). Station numbers along the transect are indicated on the X
934 axis. Main longitudes (E and W) are indicated above.

935

936 **Figure 7.** Correlations between the a) Trichome concentration estimated from PE > 10 µm or
937 Chla(Tri) and the FTL$_{Trichodesmium}$ abundance (colony counts by UVP5) (N L$^{-1}$) b) Chla
938 (Trich.) vs Trichome concentration from visual counts (N L$^{-1}$).

939

940 **Figure 8.** IOPS: a) Backscattering spectrum (log (b$_{bp}$ (m$^{-1}$)) vs log (wavelength) measured
941 by a HOBILABS Hydroscat-6 in *Trichodesmum* rich waters showing troughs at the
942 absorption wavelengths (in red) and at an oceanic station of the Diapalis 2001-2003 data
943 basis with the same H6, c) Section from 0-150m of log(b$_{bp}$ (555)). Ocean Data View sections
944 Schlitzer, R., Ocean Data View, http://odv.awi.de, 2016.

945

946 **Figure 9.** IOPS (continued): a) *In situ* absorption spectrum of *Trichodesmum* rich waters as
947 measured by the filter technique showing MAA's absorption at 330 and 360 nm wavelengths
948 and b) idem for low *Trichodesmium*, c) OUTPACE section of a$_P$(330) (upper panel), and
949 a$_P$(442) (lower panel). Ocean Data View sections Schlitzer, R., Ocean Data View,
950 http://odv.awi.de, 2016.

951

952 **Figure 10.** a) Relationship (Log/Log) between a$_P$ (330) and the FTL$_{Trichodesmium}$ abundance
953 (colony counts by UVP5) (N.m$^{-3}$) at all station/ depths (0-150m) b) Vertical distributions of
954 a$_P$(330)/a$_P$(676) at all stations, c) OUTPACE sections from 0-150m of the surface ratio
955 a$_P$(330)/a$_P$(676), and trichome concentration (visual counts) along the transect. Station
956 numbers along the transect are indicated on the X axis. Main longitudes (E and W) are
957 indicated above.


959 **Figure 11.** Correlations between the Chla (fluorimetry) and the ratio of nL$_w$(λ)/nL$_w$(565 nm)

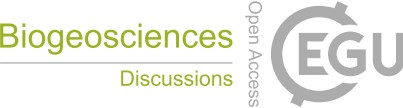



at different UV and visible wavelengths. Equations and determination coefficient ($r^2$) of the
power law are indicated for each wavelength a) 305, b) 325, c) 340, d) 380, e) 412, f) 443,
and g) 490 nm). All stations of the OUTPACE and BIOSOPE transect are reported. In black,
OUTPACE, in blue, BIOSOPE

**Figure 12.** Principal component analysis (PCA), based on Pearson's correlation matrices,
computed on the $nL_w(\lambda)$ and TChla for OUTPACE (a, b) and for BIOSOPE ) (c, d). For
OUTPACE (a,b) all surface data were used, including 7 days at LDA, LDB, LDC (n=37). For
BIOSOPE, all surface data (n = 17) were used (c,d).





**APPENDIX A: AOPS measurements and processing**

For in-water sensors, the Full- Width Half-Maximum (FWHM) of the channels was 2 nm for 305, 325 and 340 nm, and 10 nm for 380, 412, 443, 490 and 565 nm. For in-air sensors, the FWHM of the channels was 2 nm for 305, 325 and 340 nm, 10 nm for 380 nm, and 20 nm for 412, 443, 490 and 565 nm. The MicroPro free-fall profiler was operated from the rear of the ship and deployed 20-30 m away to minimize the shadowing effects and disturbances of the ship. Surface irradiance ($Es(\lambda)$, in $\mu W\ cm^{-2}\ nm^{-1}$), which is equivalent to the downward irradiance just above the sea surface ($Ed(0+, \lambda)$), was simultaneously measured at the same channels on the ship deck using other OCR-504 sensors to account for the variations of cloud conditions during the cast. Details of cast measurements are as follows. Rejection was the case at SD6 ($2^{nd}$ profile), during the long duration stations LDC ($2^{nd}$ profile day1, $2^{nd}$ profile day2, $1^{st}$ profile day3, $2^{nd}$ profile day5) and LDA (1st profile DAY5), LDB ($2^{nd}$ profile DAY3) an LDC ($2^{nd}$ profile DAY1, $2^{nd}$ profile DAY2, $2^{nd}$ profile DAY5). In total, all stations were characterized by at least 1, 2 profiles and sometimes 3 profiles. Only 2 values of $nLw(\lambda)$ at 305 nm (SD5 and SD14) showed some suspicious radiometric values among the 30 nLw profiles.

$Ed(\lambda)$ was taken from the OCR Hyperpro values from 400 to 700 nm and then integrated using the formula (Tedetti et al., 2007, eq. 1) where $Ed,_{PAR(Z)}$ is the downward irradiance in the spectral range of PAR at depth Z (quanta $cm^{-2}\ s^{-1}$), $\lambda$ is the wavelength (nm), h is the Planck's constant (6.63.10-34 J s), c is the speed of light in the vacuum (3.108 m s-1) and $Ed(Z, \lambda)$ is the downward irradiance at depth Z (mW $cm^{-2}\ nm^{-1}$). Downward attenuation coefficient was determined in accordance with their eq. 2, where $Ed(0-,\lambda)$ is the downward irradiance beneath the surface. Because of the wave-focusing effects leading to fluctuations in in-water irradiance near the surface, irradiance data of the first meters were omitted from the calculation and $Ed(0-,\lambda)$ was theoretically computed from deck measurements as in their eq. 3, where alpha (0.043) is the Fresnel reflection albedo for irradiance from sun and sky.

The diffuse attenuation coefficient for upward irradiance was determined from the slope of the linear regression of the log-transformed upward radiance versus depth in accordance with the equation between $Lu(Z1, \lambda)$ and $Lu(Z2, \lambda)$ the upward radiances ($\mu W\ cm^{-2}\ sr^{-1}$) at depths Z1 and Z2 (m), respectively (Tedetti et al., 2010). As for $Kd(\lambda)$, the depth





interval within the upper water column used for the KL($\lambda$) determination was chosen from a
visual examination of each log-transformed profile and was typically 5, 10, 15, 20, or 30 m,
depending on the stations and wave bands. The determination coefficients ($r_2$) of the KL($\lambda$)
calculation were >0.98. Water-leaving radiance (Lw($\lambda$) in $\mu W\ cm^{-2}\ sr^{-1}$) was then derived
(their equation 2) where Lu(0-, $\lambda$) is the upward radiance beneath the sea surface computed by
extrapolating Lu(Z, $\lambda$) to the sea surface from KL($\lambda$) and equation (1), t (0.975) is the upward
Fresnel transmittance of the air-sea interface, and n (1.34) is the refractive index of water.
Normalized water-leaving radiance (nLw($\lambda$) in $\mu W\ cm^{-2}\ sr^{-1}$) was determined by the formula
(equation 3) by dividing  the water-leaving radiance (Lw($\lambda$) by Es($\lambda$) the surface irradiance
and multiplying by F0($\lambda$) the solar irradiance at the top of the atmosphere, at the mean Earth-
Sun distance ($mW\ cm^{-2}$).  F0($\lambda$)  data in the ranges 305 –340 nm and 380 – 565 nm were used
from Thuillier et al. [1997, 1998], respectively as in Tedetti et al. (2010).

**APPENDIX B**
Table 1. Pump sampling between SD3 and SD4 in the Melanesian Archipelago for calibrating
the SIMBADA instrument during the OUTPACE cruise. HPLC from NASA.

| SIMBADA survey | Longitude (E) | Latitude (S) | TChla-NASA (mg.m-3) |
|---|---|---|---|
| Surf 1  4/03/16 | 166.978 | -19.704 | 0.5785 |
| Surf 2 4/03/16 | 166.6956 | -19.837 | 0.384 |
| Surf 3  4/03/15 | 166.696 | -19.847 | 0.357 |
| Surf 4 4/3/15 | 166.779 | -19.869 | 0.3135 |
| Surf5 4/03/15 | 166.956 | -19.896 | 0.3185 |
| Surf 6 4/03/15 | 167.167 | -19.923 | 0.397 |
| Surf 7 4/03/15 | 167.383 | -19.955 | 0.326 |
| Surf 8 4/03/15 | 167.639 | -19.977 | 0.2615 |
| Surf 9 5/03/15 | 167.817 | -21.447 | 0.269 |
| Surf 10 6/03/15 | 169.445 | -21.497 | 0.19 |















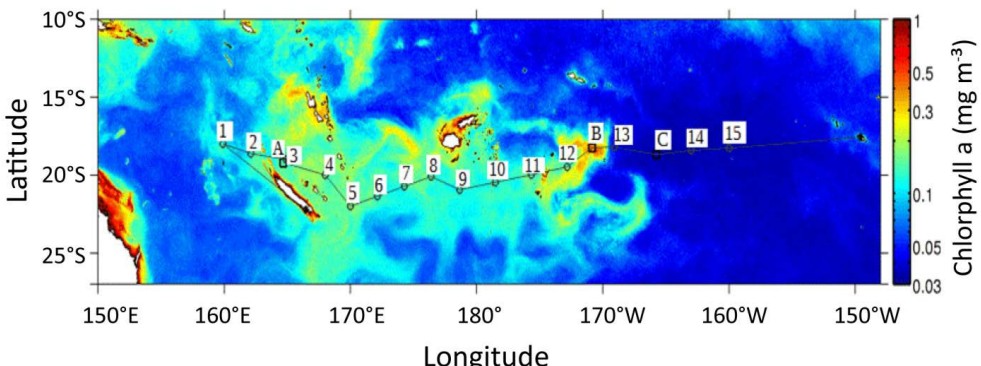











**Fig. 1**








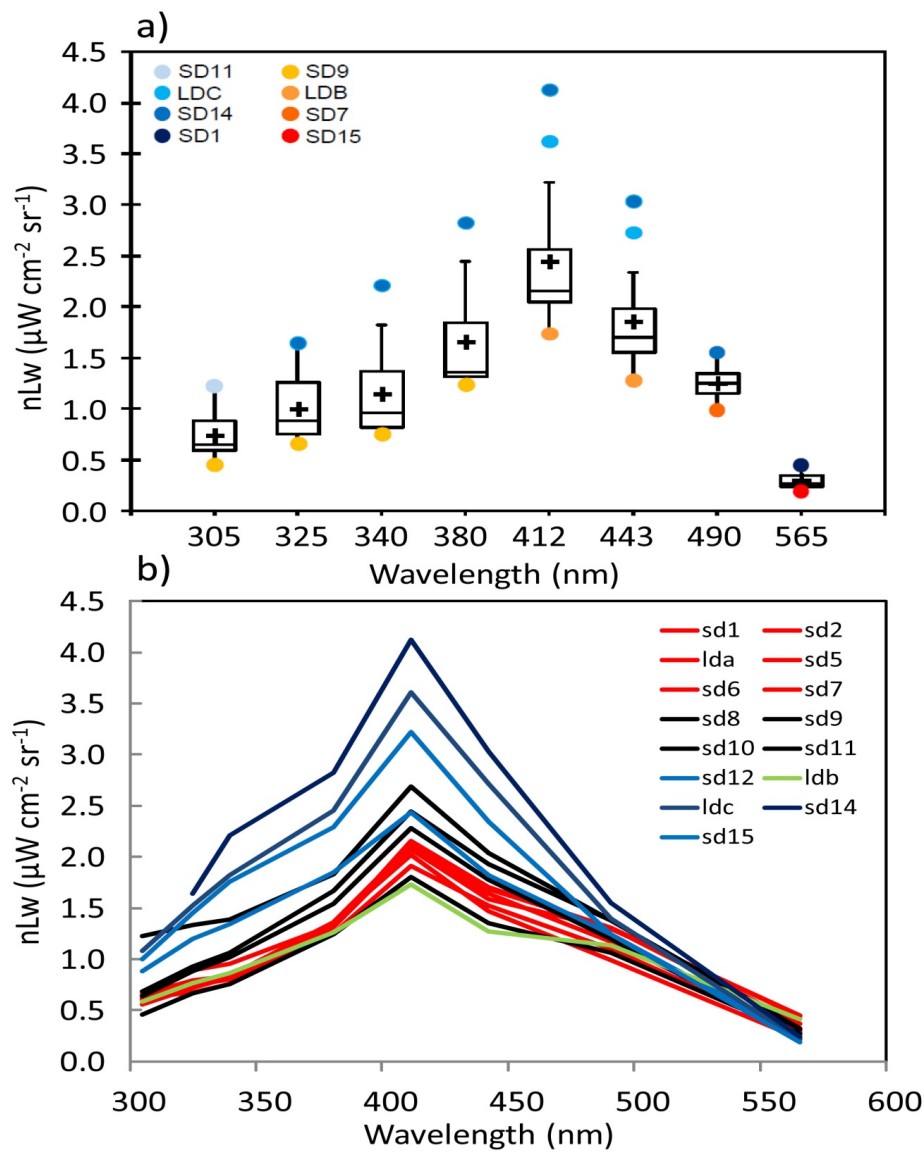



**Fig. 2**







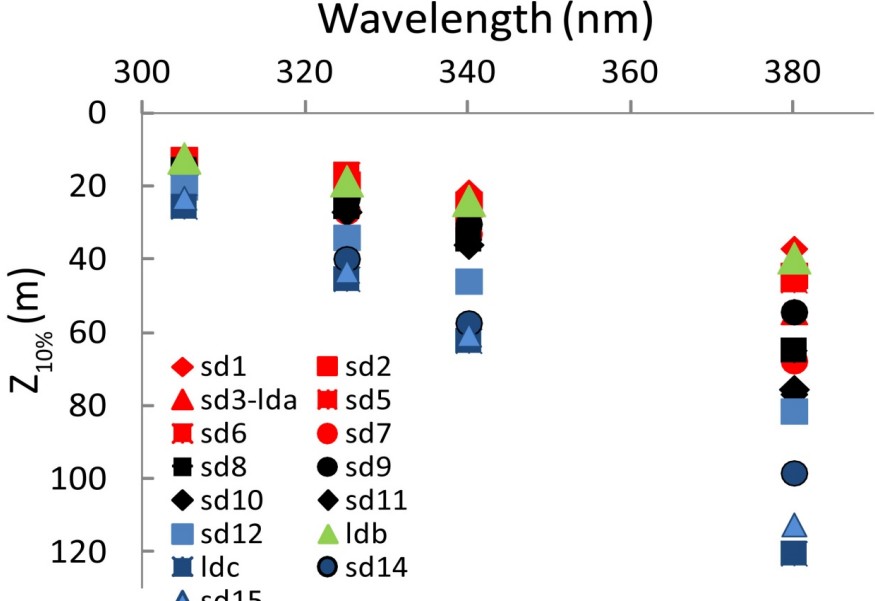










**Fig. 3**





**Fig. 4**



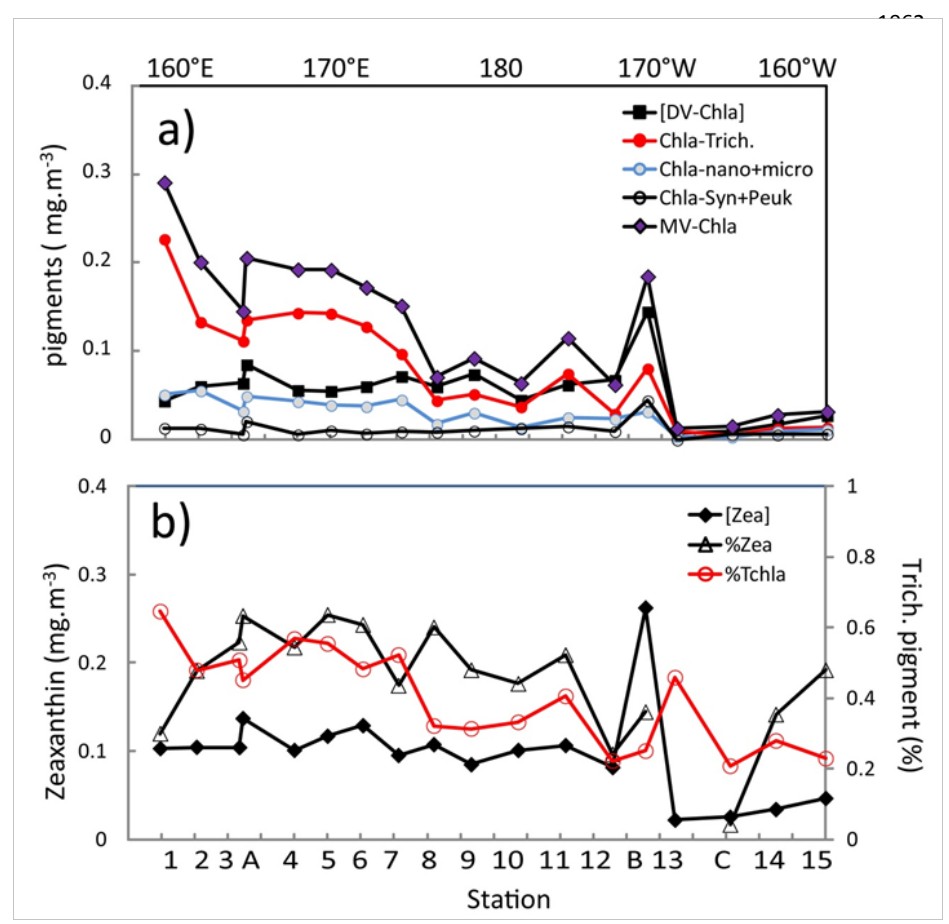





**Fig. 5**









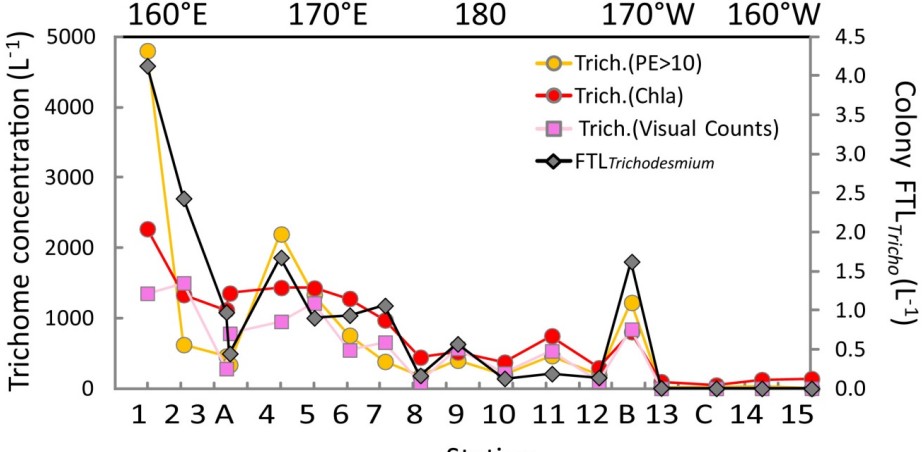








**Fig. 6**










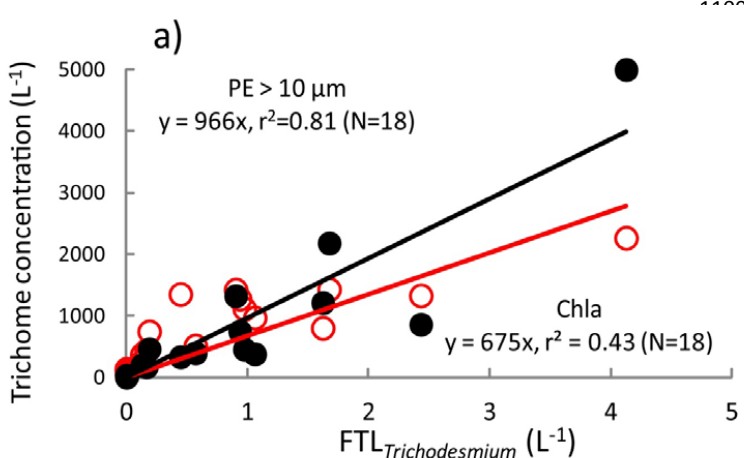

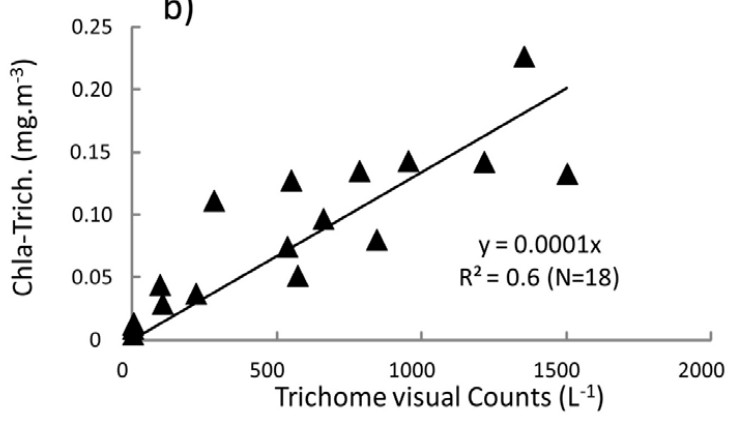






**Fig. 7**







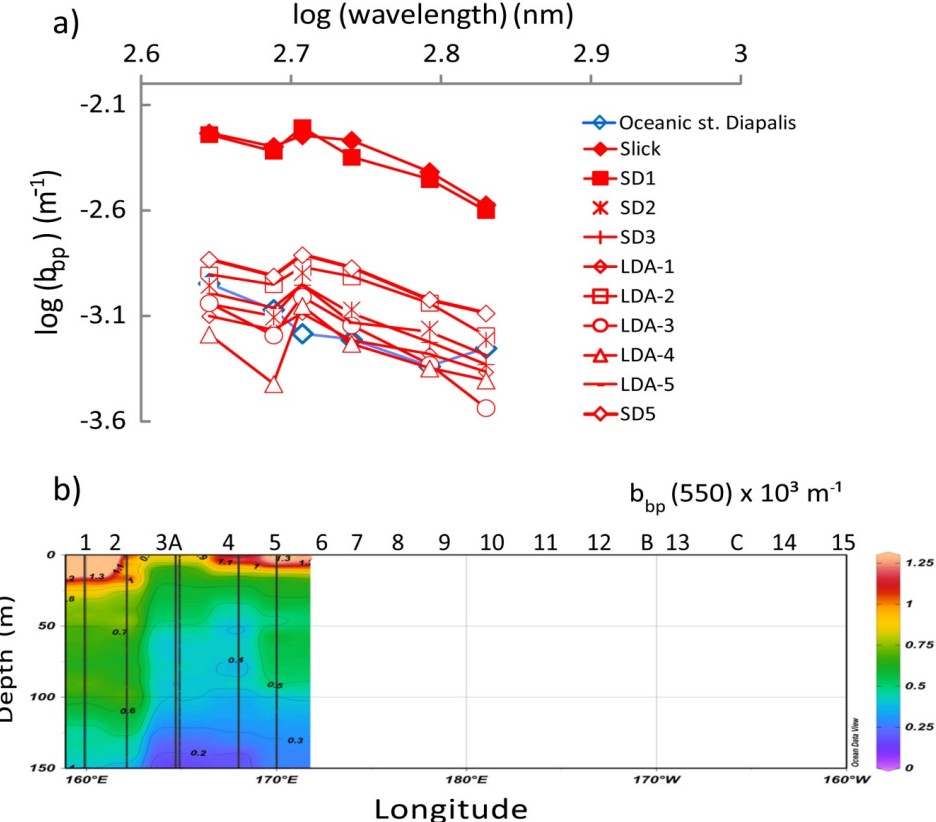


**Fig. 8**





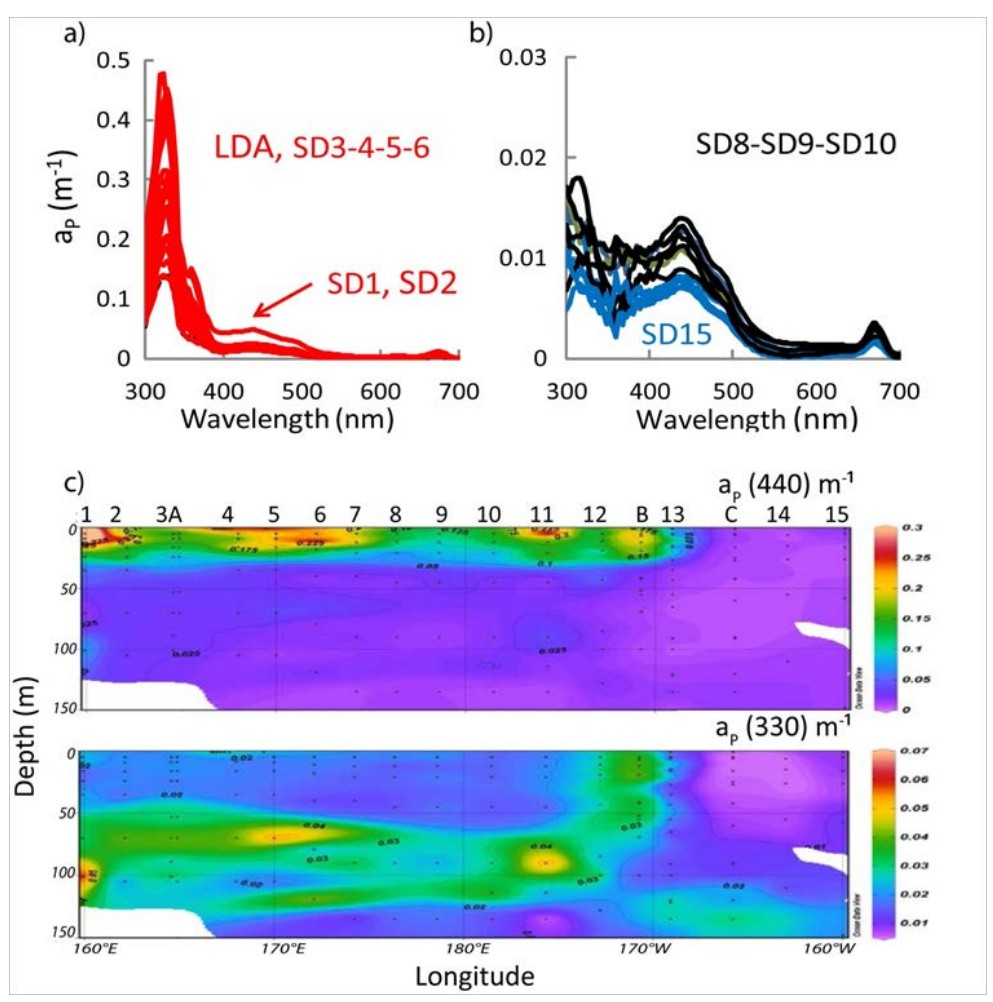




**Fig. 9**





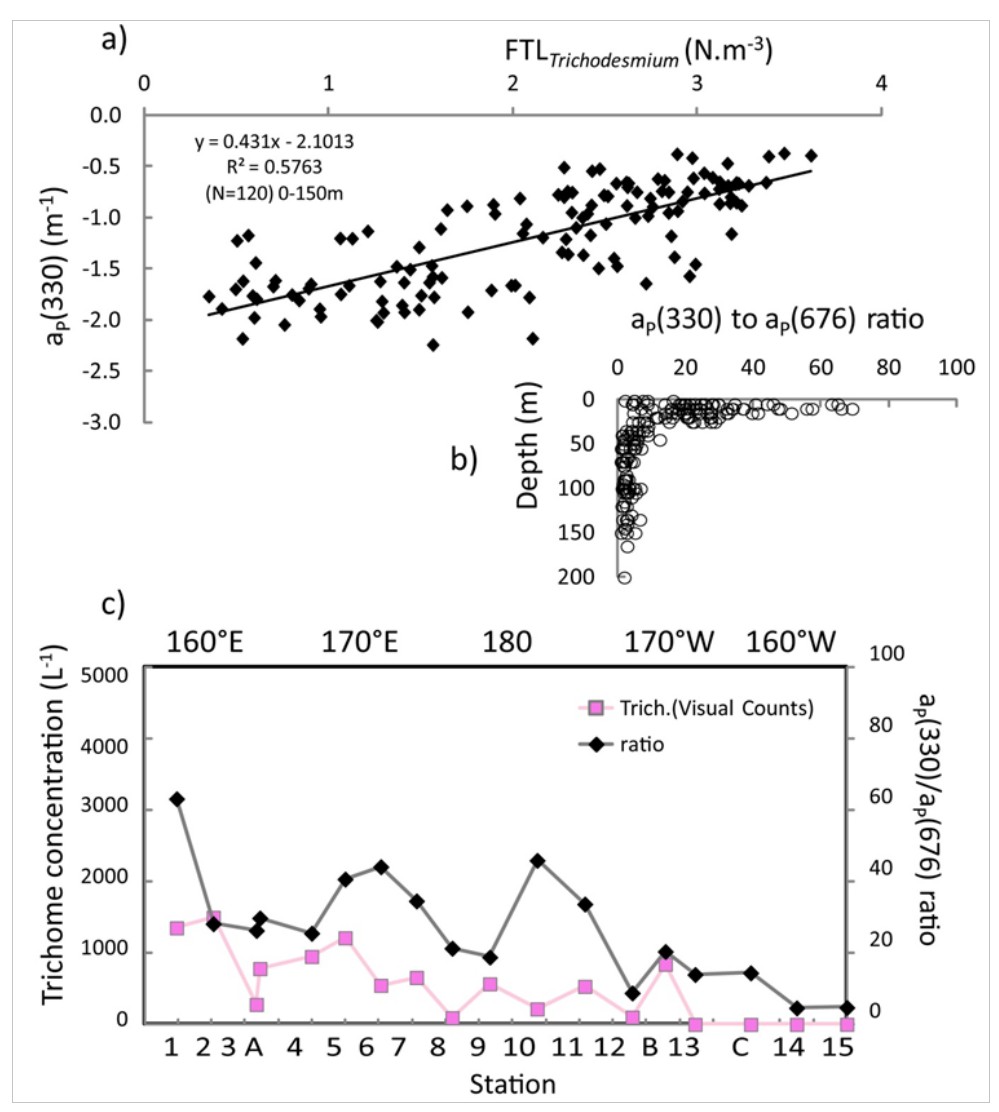



**Fig. 10**







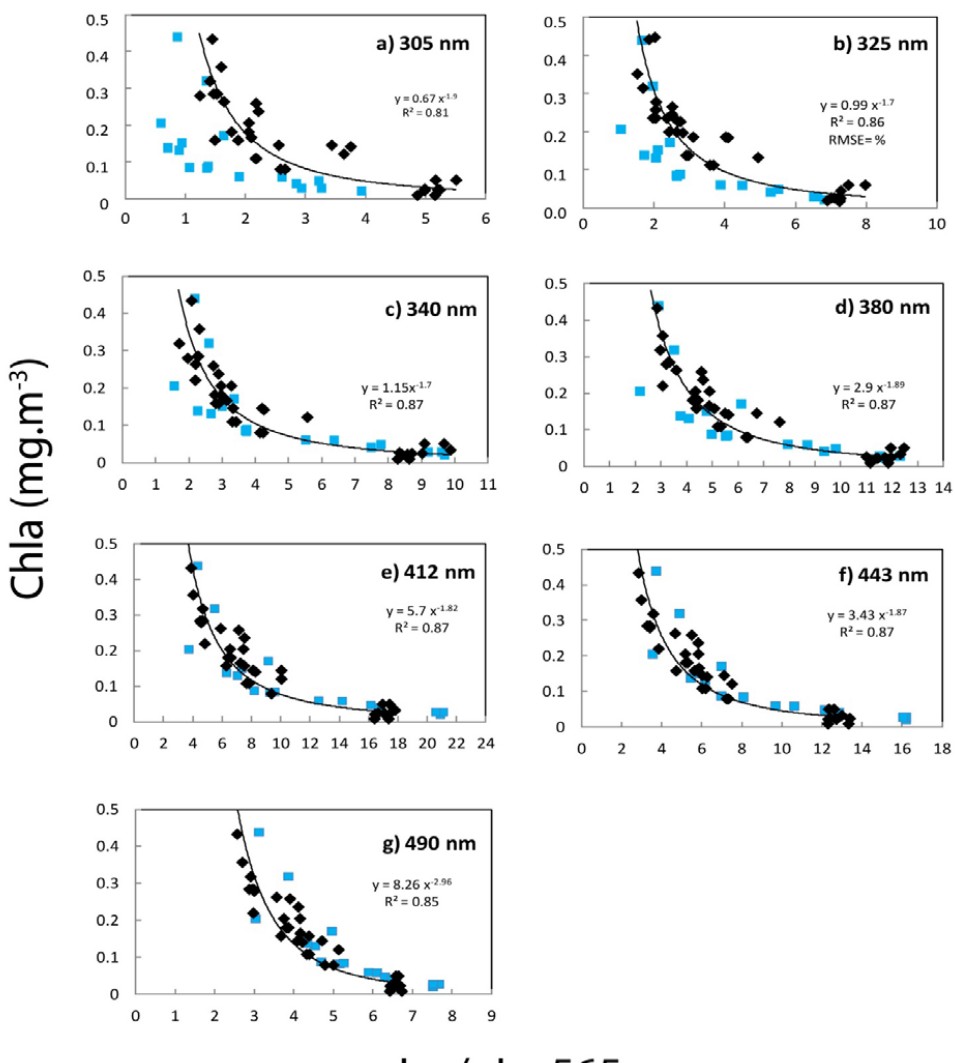


**Fig. 11**



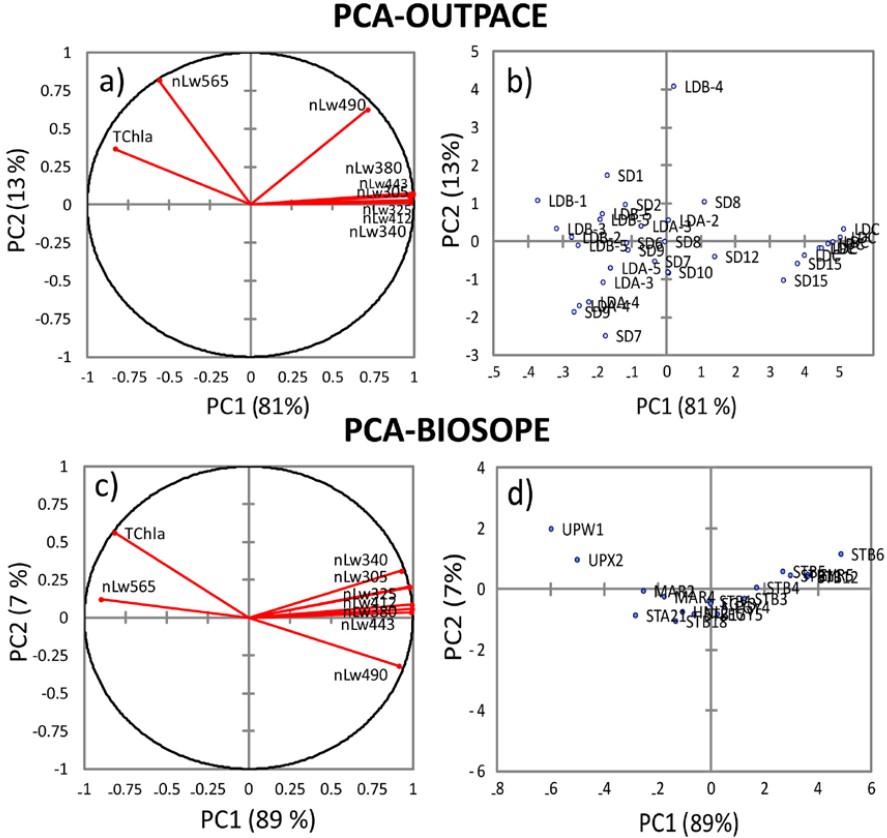







**Fig. 12**