# Peer review of "Diazotrophic *Trichodesmium* impact on UV-Vis radiance and pigment composition in the South West tropical Pacific"

_Biogeosciences, 2017_

## Referee Comment (RC1) · Anonymous Referee #1 · 15 Feb 2018

General comments: Trichodesmium and optical properties were investigated by comprehensive observation of the South West tropical Pacific waters. This paper seems to have some halfway conclusions, but gives valuable information to help future progress of the understanding of Trichodesmium bloom and its monitoring.

Specific comments:

Line 53-54: "Trichodesmium detection should 53 then involve examination of nLw at the green and yellow wavelengths.": 490nm is in between blue and green (greenish blue), and 565nm is in between green and yellow (yellowish green).

Line 411-412, "It showed large troughs due to absorption maxima at these wavelengths

in the blue channel (Fig. 6a-d).": "(Fig. 6a-d)" is wrong?

Line 455-456, "83% of total variance": 81% in fig. 12 a).

Line 459, "PC2 represents 9.4% of the total variance." 13% in fig. 12 a).

Line 467, "only 5% for PC2 (Fig. 12c).": 7% in fig. 12 c).

line 509-511: The sentence seems to be duplicated.

Fig.1, "Chlorophyll composite from MODIS on the period of the OUTPACE cruise. The positions of the short (long) duration stations are shown by cross (plus) symbols.": I could not see the "cross (plus) symbols."

Fig. 4: a)∼d) are not shown

Fig. 7 a): Please explain the colors of black and red.

Fig.8: Please show wavelength on the axis instead of log (log10?) values.

Fig. 9, "a) In situ absorption spectrum of Trichodesmum rich waters as measured by the filter technique showing MAA's absorption at 330 and 360 nm wavelengths": I could not see "MAA's absorption at 330 and 360 nm" in this figure.

[Figure]

---

## Referee Comment (RC2) · Anonymous Referee #2 · 12 Mar 2018

**General comments**

Throughout the review, I use (Y) to refer to line Y of the print version of the discussion paper.

This paper examines the distribution of *Trichodesmium* along a transect in the SW Pacific using pigment and camera data, and provides accompanying optical measurements that are related to ocean color. *Trichodesmium* abundance along the transect is described, and statistical analyses relating variability in water-leaving radiance relative to changes in chlorophyll concentration are provided. The authors conclude that certain spectral regions potentially influenced by the presence of *Trichodesmium* are good

candidates for detection and quantification of this species from ocean color.

I believe this is a valuable dataset with concurrent measurements of phytoplankton community composition and optical properties of seawater. Such datasets are needed to advance algorithm development for remote-sensing of specific functional types, as well as to provide insight into the performance and limitations of more general algorithms (e.g., Chl, POC). The demonstration and general concurrence of multiple techniques to estimate *Trichodesmium* abundance is useful, and provides a nice description of changes in community composition along the 4000-km transect and across frontal features.

I was disappointed, however, in the Discussion section of the paper. Most of the Discussion sections are very short, generally reiterate basic ideas from the literature, and call for more research. There are almost no real new concepts or conclusions given. Furthermore, a major goal of the paper (based on title and abstract) is to describe the influence of *Trichodesmium* abundance on ocean color, and this appears to be addressed only to a small extent and in a more or less qualitative way. The authors present some evidence on the influence of this species on IOPs (e.g., increased absorption coefficients in some bands, increased particulate backscattering), yet in the end their PC analyses only examines differences in nLw vs. Chl relationships and compares it to data from the S. Pacific Gyre, and then speculate that the differences in a few bands are likely due to these IOPs (or phycoerythin fluorescence contributions). With all the measurements conducted by the authors, it was disappointing that they state that "more work is needed" and then do not perform any analyses (even simple optical modeling) to confirm that the changes in IOPs they relate to *Trichodesmium* abundance have a measureable influence on water-leaving radiance that is consistent with their observations. What is the point of collecting and presenting results from all these measurements if they are not used in any quantitative sense?

Additionally, there are multiple existing algorithms (cited in the paper) for estimating *Trichodesmium* abundance from ocean color. It seems that the authors' dataset

represents a good opportunity to test such algorithms with in situ data and provide some indication of how well (or not) these algorithms perform. I am not sure why this was not done, but it would help to provide some definitive conclusions and useful outcomes from the study.

**Specific comments**

(45): "LDB" has not been defined or described, so the use of it here is confusing.

(146): Since the optical depth interval depends greatly on wavelength, which spectral band was used to calculate the integrated concentration? Or was the depth interval varied for each wavelength?

(156): -80C is not the temperature of liquid nitrogen

(192): I assume you mean >, not <, 200 um?

(274): The description of the pathlength amplification correction is missing.

(377): What is the depth sampled by the "pump" samples?

(412): I assume you mean Fig. 9a-d?

(420-424): I have a hard time following the description of Fig. 9 results. First, it appears that the labels in Fig. 9c are reversed (i.e., ap(330) should be the upper panel, ap(440) the bottom)? Second, I don't understand the references to 350 and 442 nm (which are not shown in the figure). Third, what is the meaning of the "(>80)" in the sentence "High values (>80) of ap(330)..."?

(451): Are the input "nLw values" the magnitudes, or have they been normalized in any way?

(476) The title of this section includes contributions to absorption, but absorption is not mentioned anywhere in the paragraph.

[Figure]

(487) Please explain what is "Diapalis".

(496) I was hoping that with the collected set of measurements this would be accomplished by this study. It is rather disappointing to read to this point, and then have this statement in which the authors basically defer on addressing the stated purpose of the paper.

(499 - 524): I do not see any point to these two sections (4.2 and 4.3). They basically reiterate observations from previous studies, and state no clear conclusions or provide new insights from this study.

(609): Earlier in the manuscript (line 427), it is stated that the MAAs index was variable and not tightly related to *Trichodesmium*. This sentence seems to contradict that statement. I do not see a figure that explicitly shows a correlation between the MAAs index and *Trichodesmium* abundance.

(906): Provide the specific concentration ranges that correspond to "high, median, and oligotrophic" Tchla values which the color-codes are based upon.

(954): It is unclear how you can have sections from 0-150m of a "surface" ratio.

(Fig. 4): The subpanel labels (a, b, ...) are not provided in the figure.

(Fig. 5): In Fig.. 5b, the right axis needs to be multiplied by 100 in order to have units of "percent".

(Fig. 9): As described earlier, it seems that labels in Fig. 9c are reversed?

**Technical corrections**
There are numerous typographical errors along with incomplete or repeated sentences throughout the text (more than I care to tabulate), and suggest that the authors carefully proofread the manuscript or ask a colleague do it.

---

## Author Comment (AC1) · 28 Apr 2018

The authors acknowledge the reviewers for the helpful comments

General comments Throughout the review, I use (Y) to refer to line Y of the print version of the discussion paper. This paper examines the distribution of Trichodesmium along a transect in the SW Pacific using pigment and camera data, and provides accompanying optical measurements that are related to ocean color. Trichodesmium abundance along the transect is described, and statistical analyses relating variability in water-leaving radiance relative to changes in chlorophyll concentration are provided. The authors conclude that certain spectral regions potentially influenced by the presence of Trichodesmium are good candidates for detection and quantification of this species from ocean color.

Answer: We thank the Reviewer "2 for these positive and constructive comments.

I believe this is a valuable dataset with concurrent measurements of phytoplankton community composition and optical properties of seawater. Such datasets are needed to advance algorithm development for remote-sensing of specific functional types, as well as to provide insight into the performance and limitations of more general algorithms (e.g., Chl, POC). The demonstration and general concurrence of multiple techniques to estimate Trichodesmium abundance is useful, and provides a nice description of changes in community composition along the 4000-km transect and across frontal features.

Answer: We thank the Reviewer "2 for these positive and constructive comments.

I was disappointed, however, in the Discussion section of the paper. Most of the Discussion sections are very short, generally reiterate basic ideas from the literature, and call for more research. There are almost no real new concepts or conclusions given.

Answer: We understand the Reviewer comments. Therefore, in the revised manuscript, we have substantially strengthened the discussion section and brought more comparison with published work on the detection of Trichodesmium by their optical signature.

Furthermore, a major goal of the paper (based on title and abstract) is to describe the influence of Trichodesmium abundance on ocean color, and this appears to be addressed only to a small extent and in a more or less qualitative way.

Answer: In the revised manuscript, we have added some paragraphs in the discussion

and some more quantitative results, in addition to ACP results; which allowed distinguishing a characteristic radiance signal at 490 and 555 nm linked to a 2nd axis (13% of total variance) during OUTPACE. This result is robust as it appears in all PCA we have done to complete the interpretation of our data.

The authors present some evidence on the influence of this species on IOPs (e.g., increased absorption coefficients in some bands, increased particulate backscattering), yet in the end their PC analyses only examines differences in nLw vs. Chl relationships and compares it to data from the S. Pacific Gyre, and then speculate that the differences in a few bands are likely due to these IOPs (or phycoerythrin fluorescence contributions).

Answer: We agree with the Reviewer that other parameters could have been presented in PCAs, as for example backscattering coefficient or particulate absorption coefficient. We could also have used the UVP5 trichodesmium abundance instead of Chla. We previously performed PCAs on OUTPACE data including Phycoerythrin concentration > 10 $\mu$m (MaxPE), zeaxanthin concentration (zea), Chla fluor, and UVP5 colony abundance in an additional PCA 1 (see Figure 1 below). It shows that Chla and UVP5 tricho abundance ends up totally correlated so for the manuscript, we only used Chla.

See joined Figure 1

Figure 1. PCA1. OUTPACE cruise only including all parameters

We also carried out a PCA on particulate absorption, aP, at all depths and all stations, with UVP5 at while aP380nm was not (and that the aP at the visible channels were not correlated with the UVP5 colony concentrations (PCA 2 (see Figure 2 below).

See joined Figure 2

Figure 2. PCA2: on particulate absorption coefficients at the same wavelengths that normalized water-leaving radiances, all stations. (Stations depths are indicated as SD1 9 for SD1 at 9m). UVP5 is at the same position as nLw at 324nm Finally, we also performed a PCA between the radiance nLw and the particulate backscattering coefficient at 550nm for OUTPACE and BIOSOPE. For OUTPACE, only stations where bbp(550) was measured (SD1 to SD6) were used, and for BIOSOPE, bbp550 was calculated from Chla as in Huot et al., 2008 from equation "bbp = $\alpha$1 [Chl]$\beta$, with coefficients established for a Hydroscat-6 by Stramski et al., 2008 for BIOSOPE) (additional PCA3, Figure 3a,b). Particulate backscattering coefficient at 550 nm is found at the same position in the PCA than Chla in our manuscript (our Fig. 12 in the manuscript).

See joined Figure 3 a) b)

Figure 3 PCA3: between bbp(550) and nLw at OUTPACE (a) and at BIOSOPE (b). bbp(550) measured with the Hydroscat-6 at OUTPACE, calculated from Huot et al., 2008 at BIOSOPE.

Conclusions of these partial PCAs : - From PCA1 Fig 1 : UVP5 (noted Moyfibsta_10m), PEmax (PE > 10 $\mu$m), zeaxanthin and Chla are linked (on the same angles on the correlation circle) and correlated with nLw565. - From PCA2 Fig 2: Absorption coefficient at all wavelengths of the interval 305-340nm are linked with UVP5. AP380nm is not linked to UVP5. AP coefficients in the visible domain are not linked with UVP5. - From PCA3 Fig 3: Particulate backscattering coefficient at 550 nm is found at the same position in the PCA than Chla, which suggests a strong relationship with Chla.

With all the measurements conducted by the authors, it was disappointing that they state that "more work is needed" and then do not perform any analyses (even simple optical modeling) to confirm that the changes in IOPs they relate to Trichodesmium abundance have a measureable influence on water-leaving radiance that is consistent with their observations.

Answer: We have corrected this by using a simple optical model relating Rrs to the bb/a ration, and using our measurements of bbp and aP when both available (SD1 to SD6) and compared results to the in situ radiances and to modeled ones obtained by Subramaniam et al. (1999) for a mix of Trichodesmium.

[Figure]

What is the point of collecting and presenting results from all these measurements if they are not used in any quantitative sense?

Answer: We agree with the Reviewer. We have added some more quantitative results in the discussion paragraph.

As the Hydroscat-6 failed at station 6, we do not have measurements of the backscattering coefficient at all stations over the whole OUTPACE transect. We have valuable measurements on Trichodesmium slicks, which can be compared to the ones obtained in tanks. Also, we used DIAPALIS data (9 cruises at 167° 20°S, 2001-2003) obtained with the same Hydroscat-6 instrument (unpublished results) in and out of the Trichodesmium slicks. CDOM spectra measurements were heavily impacted by MAA's peaks in dissolved absorption in the Western part of the transect. We think that these spectra have first to be corrected from the MAA's influence at least from SD1 to SD6 to be used in the statistical analysis.

Additionally, there are multiple existing algorithms (cited in the paper) for estimating Trichodesmium abundance from ocean color. It seems that the authors' dataset represents a good opportunity to test such algorithms with in situ data and provide some indication of how well (or not) these algorithms perform.

Answer: We have added a discussion paragraph on this subject. In the revised version, we have discussed about the rationale of the results given by the PCA on the normalized water radiance of OUTPACE in comparison of conclusions published previously on the possibility of detection of Trichodesmium on nLw (Subramaniam et al., 1999; applied by Westberry et al., 2005; 2006) whose model normalized water radiances as empirically determined on pure or mixture of Trichodesmium rich waters and as a function of Chla, and specific Trichodesmium IOPs could be used as a comparison (Subramaniam et al., 2002). In these models, the fluorescence of the PE was included as it probably impacts the 565 nm radiance. Moreover, in situ radiance obtained on a Trichodesmium blooms on the Easteen US coast have also been used (Subramaniam

et al. 2002).

I am not sure why this was not done, but it would help to provide some definitive conclusions and useful outcomes from the study.

Answer: Please note that for surface Trichodesmium slicks and mats detection, a companion paper is proposed to the Special Biogeoscience, which addresses the specific case of surface slicks on MODIS radiances in the NIR part of the spectrum (Rousset et al., in revision). The radiometric measurements we undertaken in the present study are representative of Trichodesmium concentration from 0 to 30 meters. Therefore, we do not address here the question of surface slicks. The other algorithms that allow to discriminate Trichodesmium at low concentrations (0.5 to 2 mg m-3) of Subramaniam et al. and used in the Westberry et al., 2005, 2006 have been compared to our approach and a discussion has been added in the revised manuscript.

Specific comments

(45): "LDB" has not been defined or described, so the use of it here is confusing.

Answer: LDB means "Long Duration station B". Description of stations can be found in Moutin et al., 2017, this issue. We corrected it in the revised manuscript

(146): Since the optical depth interval depends greatly on wavelength, which spectral band was used to calculate the integrated concentration? Or was the depth interval varied for each wavelength?

Answer: We agree with the Reviewer. The depth interval within the upper water column used for the $KL(\lambda)$ determination or nLw values was chosen from a visual examination of each log-transformed profile and was typically 10, 15, 20, or 30 m, depending on the stations and wave bands.

(156): -80C is not the temperature of liquid nitrogen Answer: Exactly. We corrected it in the revised manuscript (-180°C)

(192): (192) I assume you mean >, not <, 200 um?

Answer: Yes, indeed it was > 200 nm (corrected)

(274): The description of the pathlength amplification correction is missing.

Answer: This description of the correction of the pathlength amplification factor used was added in the text. The pathlength amplification factor ($\beta$) due to filter multiple scattering was corrected with the coefficients of Mitchell et al., 1990. The Optical density of the equivalent suspension, ODs, was obtained from the value on filter, ODf, by the formula ODs= A ODf + B (ODf)2 . We took the A and B coefficients determined by Mitchell et al. 1990 which were well suited for the oligo- to mesotrophic waters in the Pacific ocean as already determined in Dupouy et al., 2010.

(377): What is the depth sampled by the "pump" samples?

Answer: The depth of the water sampled by the continuous Pump system installed on the Atalante was 3.5 meters. This allowed to sample Trichodesmium surface slicks (as seen on different figures of the paper). (412): I assume you mean Fig. 9a-d?

Answer: It is Figure 8 ab (Backscattering description)

(420-424): I have a hard time following the description of Fig. 9 results. First, it appears that the labels in Fig. 9c are reversed (i.e., ap(330) should be the upper panel, ap(440) the bottom)?

Answer: We corrected it. This inversion was unfortunate and we are sincerely sorry for this. Of course, ap(330) was the upper panel and the ap(440) the lower panel.

Second, I don't understand the references to 350 and 442 nm (which are not shown in the figure).

Answer: This was corrected to 330 nm and 440 nm in order to harmonize the text and figures.

Third, what is the meaning of the "(>80)" in the sentence "High values (>80) of ap(330)..."?

Answer: Sorry we corrected the mistake in the revised manuscript. This value is the one of the ratio ap330/ap676 and not the ap330nm.

(451): Are the input "nLw values" the magnitudes, or have they been normalized in any way?

Answer: Yes, nLw values have been normalized. As described in the Appendix, Normalized water-leaving radiance (nLw($\lambda$) (in $\mu$W cm-2 sr-1) was determined by the formula (equation 3 in Tedetti et al., 2010) by dividing the water-leaving radiance (Lw($\lambda$) ($\mu$W cm-2 sr-1) by Es($\lambda$) ($\mu$W cm-2) the surface irradiance and multiplying by F0($\lambda$) the solar irradiance at the top of the atmosphere, at the mean Earth-Sun distance ($\mu$W cm-2).

(476) The title of this section includes contributions to absorption, but absorption is not mentioned anywhere in the paragraph.

Answer: We corrected this by discussing also absorption results by comparison with literature. (487) Please explain what is "Diapalis".

Answer: The explanation of "Diapalis" was "DIAzotrophy in the PACific on the ALIS ship" (definition now included in the revised manuscript).

(496) I was hoping that with the collected set of measurements this would be accomplished by this study. It is rather disappointing to read to this point, and then have this statement in which the authors basically defer on addressing the stated purpose of the paper.

Answer: Right. We have now added a discussion section about the impact of IOPs characteristics on radiance levels.

(499 - 524): I do not see any point to these two sections (4.2 and 4.3). They basically

reiterate observations from previous studies, and state no clear conclusions or provide new insights from this study.

Answer: We agree with the Reviewer. In the revised manuscript, we have entirely changed the discussion by comparing our results to other measurements of radiance on Trichodesmium patches.

(609): Earlier in the manuscript (line 427), it is stated that the MAAs index was variable and not tightly related to Trichodesmium. This sentence seems to contradict that statement. I do not see a figure that explicitly shows a correlation between the MAAs index and Trichodesmium abundance.

Answer: Our sentence (line 427) referred to the surface only (15 samples). For this layer, at some stations, some low UVP5 concentrations were sometimes associated with a high MAA index. This was the case at SD 5, 6, 7. This was due to uncertainties in the UVP5 abundance as we checked that all the spectra exhibit the double peak at 330 and 360nm typical of Trichodesmium. Nevertheless, the absorption spectrum of SD10 exhibited a reduced second peak at 360nm, indicating the possible influence of another type of MAA's with a single peak associated to other phytoplankton group (as in Bricaud et al., 2010), associated with Trichodesmium (high MAA index with low UVP5 abundance of SD10 at Fig 10c). Nevertheless, when considering the whole water column (all depths from 0 to 150 m, a significant correlation was found between UVP5 colony counts and the value of aP330 (our figure 10a) (same result is found for the aP330/676 ratio). AP330 and aP360 are both the wavelengths peaks of the MAA's of Trichodesmium (Dupouy et al., 2008, JARS). This was confirmed by our PCA2 on aP at all wavelengths of the Satlantic (see above) Figure 2, which showed that aP at 304, 328, 340nm were linked to UVP5, while aP at 380 nm is not. At the opposite, the aP coefficients at visible wavelengths are not linked to UVP5 concentrations. On the correlation circle, UVP5 tricho like abundance is strongly correlated with aP328 (or aP330)

[Figure]

(906): Provide the specific concentration ranges that correspond to "high, median, and oligotrophic" Tchla values which the color-codes are based upon.

Answer: Color-code of nLw spectra is as follows: Blue spectra = oligotrophic waters: TChla < 0.06 mg m-3, i.e. SD14 to SD15 including LDC; black spectra= 0.06 < TChla < 0.18 mg m-3, i.e. SD8 to SD12 around Fiji Islands, red spectra= Melanesian archipelago: 0.185 < TChla < 0.35 mg m-3, i.e. SD1 to SD7. Chla concentrations can be found at Table 1 in Annex 1. LDB was highlighted in green as it has the lowest nLw associated with a high TChla concentration (0.32 mg m-3). This was mentioned in the new legend of the Figure.

(954): It is unclear how you can have sections from 0-150m of a "surface" ratio.

Answer: We agree with the Reviewer, we corrected the legend in the revised manuscript.

(Fig. 4): The subpanel labels (a, b, ...) are not provided in the figure.

Answer: This has been corrected in the revised manuscript, by adding labels a)b)c)d)

(Fig. 5): In Fig.. 5b, the right axis needs to be multiplied by 100 in order to have units of "percent".

Answer: Right. This has been corrected and greatly helps the figure.

(Fig. 9): As described earlier, it seems that labels in Fig. 9c are reversed?

Answer: Yes, it was unfortunately reversed at the last print version of the figure. We have corrected it in the revised manuscript.

Technical corrections

There are numerous typographical errors along with incomplete or repeated sentences throughout the text (more than I care to tabulate), and suggest that the authors carefully proofread the manuscript or ask a colleague do it.

Answer: This has been corrected in the revised manuscript. We thank you for these comments.

Please also note the supplement to this comment:
https://www.biogeosciences-discuss.net/bg-2017-570/bg-2017-570-AC1-supplement.pdf

————————————————————

**Figures des reponses**

**Fig. 1.**

[Figure]

[Figure]

**Corrected Fig. 8**

**Fig. 2.**

---

## Author Comment (AC2) · 28 Apr 2018

Anonymous Referee #1 General comments: Trichodesmium and optical properties were investigated by com- prehensive observation of the South West tropical Pacific waters. This paper seems to have some halfway conclusions, but gives valuable information to help future progress of the un- derstanding of Trichodesmium bloom and its monitoring.

[Figure]

Answer: We agree with the Reviewer. Hence, in the revised manuscript we strengthened our conclusions :

- We found that 60% of Chla and 55% of zeaxanthin was attributed to Trichodesmiumin the western pat of the transect, that the UVP5 provided a true Trichodesmium abundance linked to the filaments abundance by a factor of aggregation of 600-900, and that the UV-Vis free fall Satlantic radiometer radiance field was influenced by the presence of the Trichodesmium colonies especially in the greenish blue and yellowish green domain. New results of additional PCAs have been included in order to improve our results of radiance anomalies. A first, we did a PCA on radiances and other parameters of OUTPACE than simple Chla, i.e. with zeaxanthin, phycoerythrin > 10 $\mu$m, and with UVP5 colony abundance. At second, we performed a PCA on the particulate absorption coefficient, Ap, vs UVP5. Last, we carried out PCAs on the backscattering coefficient at 550nm and radiances for both OUTPACE and BIOSOPE which showed the correlation between the radiance at 565nm and bbp(550). Moreover, we compared the anomalies in greenish blue and yellowish green radiance indicated by the PCA, with results of empirically modeled radiance found in the literature (Subramaniam et al., 1999, 2002).

Specific comments: Line 53-54: "Trichodesmium detection should then involve examination of nLw at the green and yellow wavelengths.": 490nm is in between blue and green (greenish blue), and 565nm is in between green and yellow (yellowish green).

Answer: We agree with the Reviewer. In the revised manuscript, we thus replaced: "blue" by "greenish blue" and "yellow" by "yellowish green".

Line 411-412, "It showed large troughs due to absorption maxima at these wavelengths at the blue channel (Fig. 6a-d).": "(Fig. 6a-d)" is wrong?

Answer: We corrected this sentence by saying: "It showed large troughs due to absorption maxima at these wavelengths, which were stronger at the blue channel". Backscattering coefficients are described at Figure 8 a) b).

Line 455-456, "81% of total variance": 81% in fig. 12 a). Answer: Corrected.

Line 459, "PC2 represents 9.4% of the total variance." 13% in fig. 12 a) Answer: It is 13%.

Line 467, "only 5% for PC2 (Fig. 12c).": 7% in fig. 12 c). Answer: It is 7%

line 509-511: The sentence seems to be duplicated. Answer: Yes indeed. In the revised manuscript we removed the duplicated sentence.

Fig.1, "Chlorophyll composite from MODIS on the period of the OUTPACE cruise. The positions of the short (long) duration stations are shown by cross (plus) symbols.": I could not see the "cross (plus) symbols." Answer: Corrected. The map does not show crosses.

Fig. 4: a) to d) are not shown Answer: In the revised manuscript, we have included the lettering of each part of Fig 4.

Fig. 7 a): Please explain the colors of black and red. Answer: We explained now in the new figure and in the legend, that the black is for the relationship between PE and UVP5 colony abundance, and the red is for the relationship between Chla and UVP5 colony abundance.

Fig.8: Please show wavelength on the axis instead of log (log10?) values. Answer: Corrected. We made the modification in the revised manuscript.

Fig. 9, "a) In situ absorption spectrum of Trichodesmum rich waters as measured by the filter technique showing MAA's absorption at 330 and 360 nm wavelengths": I could not see "MAA's absorption at 330 and 360 nm" in this figure.

Answer: We are sorry for this mistake (legends have been inverted in the final version of the figure). Indeed, aP(330) is on the upper panel and aP(440) is in the lower panel. This is now corrected.

Please also note the supplement to this comment:
https://www.biogeosciences-discuss.net/bg-2017-570/bg-2017-570-AC2-supplement.pdf

—————————————————————

Interactive comment on
"Diazotrophic Trichodesmium influence on ocean color
and pigment composition in the South West tropical
Pacific"
by
Cécile Dupouy et al.

Anonymous Referee #1

General comments: Trichodesmium and optical properties were investigated by comprehensive observation of the South West tropical Pacific waters. This paper seems to have some halfway conclusions, but gives valuable information to help future progress of the understanding of Trichodesmium bloom and its monitoring.

**Answer: We agree with the Reviewer. Hence, in the revised manuscript we strengthened our conclusions :**

**- We found that 60% of Chla and 55% of zeaxanthin was attributed to Trichodesmiumin the western pat of the transect, that the UVP5 provided a true Trichodesmium abundance  linked to the filaments abundance by a factor of aggregation of 600-900, and that the UV-Vis free fall Satlantic radiometer radiance field was influenced by the presence of the Trichodesmium colonies especially in the greenish blue and yellowish green domain.**
**New results of additional PCAs have been included in order to improve our results of radiance anomalies. A first, we did a PCA on radiances and other parameters of OUTPACE than simple Chla, i.e. with zeaxanthin, phycoerythrin > 10 µm, and with UVP5 colony abundance. At second, we performed  a PCA on the particulate absorption coefficient, Ap, vs UVP5. Last, we carried out PCAs on the backscattering coefficient at 550nm and radiances for both OUTPACE and BIOSOPE which showed the correlation between the radiance at 565nm and bbp(550). Moreover, we compared the anomalies in greenish blue and yellowish green radiance indicated by the PCA, with results of empirically modeled radiance found in the literature (Subramaniam et al., 1999, 2002).**

Specific comments:
Line 53-54: "Trichodesmium detection should  then involve examination of nLw at the green and yellow wavelengths.": 490nm is in between blue and green (greenish blue), and 565nm is in between green and yellow (yellowish green).

**Answer: We agree with the Reviewer. In the revised manuscript, we thus replaced: "blue" by "greenish blue" and "yellow" by "yellowish green".**

Line 411-412, "It showed large troughs due to absorption maxima at these wavelengths at the blue channel (Fig. 6a-d).": "(Fig. 6a-d)" is wrong?

**Answer: We corrected this sentence by saying: "It showed large troughs due to absorption maxima at these wavelengths, which were stronger at the blue channel". Backscattering coefficients are described at Figure 8 a) b).**

Line 455-456, "81% of total variance": 81% in fig. 12 a).

**Fig. 1.**

[Figure]

**Corrected Fig. 8**

**Fig. 2.**

---

## Author Response (AR2)

Suggestions for revision or reasons for rejection (will be published if the paper is accepted for final publication)

Journal: Biogeosciences Discuss.

Title: Diazotrophic Trichodesmium impact on UV-Vis radiance and pigment composition in the South West tropical Pacific

Author(s): Dupouy et al.

MS No.: 2017-570

MS Type: Research Article

\noindent{\bf General comments}\\

The authors have satisfactorily addressed my comments from the initial review, and I thank them for making this effort. In my opinion the revised manuscript is much improved and better conveys the results of their study.

I have only one minor comment and a few technical corrections on the revised manuscript, all easily addressed, which the authors may wish to consider for the final version.

\noindent{\bf Specific comments}\\

Line 407: I would say the fits of nLw vs. Chla at 305 and 325 nm from OUTPACE are different from BIOSOPE, but are they necessarily "better"? Can you provide a reason why the relationships are different in these bands but not other UV bands (e.g., 340, 380 nm)? Is it related to the presence of Trichodesmium, or some other constituent covarying with Chl?

We have added a new paragraph in the text to explain these results. In the present study, Chla was well correlated to all nLw (λ) ratios [nLw (λ)/nLw (565)] with $r^2$ varying from 0.79 to 0.83 (power regressions) with RMSE (not shown) ranging from from 51 to 30% from 305 to 490 nm for OUTPACE and from 36 to 23% for BIOSOPE) according to wavelength considered (Fig. 11). The relationships between nLw (λ) and Chla were different at OUTPACE than at BIOSOPE (Fig. 11). These good relationships obtained even in the UV domain, where Chla though absorbing at 380 nm does not show any absorption peak in the UV domain, were already observed in the South East Pacific during BIOSOPE cruise, for equivalent ranges, and attributed to the fact that CDM substances

absorbing in the UV domain covary with Chla (Tedetti et al., 2010). The reason why the relationships are different at 305 and 325 nm wavelengths but not other UV bands (e.g., 340, 380 nm) is probably related to the presence of *Trichodesmium* and other constituent covarying with

10 Chl and absorbing more at 305 and 325 nm than at longer UV wavelengths. It can be noted than for the same Chla, ratios are higher at OUTPACE than at BIOSOPE, i.e. absorption would be lower in the 305 and 325 nm bands and this difference is stronger at high Chla (rich stations in the upwelling at BIOSOPE, MA stations and LDB at OUTPACE). One possible reason is that CDOM and CDM may be higher in the coastal upwelling or Marquesias waters than in the Trichodesmium rich-waters of OUTPACE.

\noindent{\bf Technical corrections}\\

Line 282: Here the surface TChla "accumulation" for station LDB is stated as 1 mg m^{-3}, yet in Table 1 the surface value is given as 0.433, the Fig. 2 legend states it is 0.42, and Line 299 again states it is 0.433. I suspect that by "accumulation" you are referring to some depth-integrated value, if that is the case please make it clear. These small inconsistencies confused this reader at least.

Corrected.

Line 329: Unless I am mistaken, Fig. 4a is not derived from HPLC pigments, as suggested by this sentence, but from the UVP measurements.

Corrected

Line 417: I realize it's rounding approximation, but the sum of the two PCs as written (81 + 13) is 94%, not 93%. The same comment applies to Line 429 (89 + 7 = 96%, not 95%).

Corrected